# ISOMETRIC REPRESENTATION LEARNING FOR DISENTANGLED LATENT SPACE OF DIFFUSION MODELS

## ABSTRACT

Diffusion models have made remarkable progress in capturing and reproducing real-world data. Despite their success and further potential, their latent space, the core of diffusion models, mostly still remains unexplored. In fact, the latent spaces of existing diffusion models still do not align close with the human perception, entangling multiple concepts in a distorted space. In this paper, we present *Isometric Diffusion*, equipping a diffusion model with isometric representation learning to better reflect human intuition and understanding of visual data. Specifically, we propose a novel loss to promote isometry of the mapping between the latent space and the data manifold, enabling a semantically and geometrically better latent space. This approach allows diffusion models to learn a more disentangled latent space, enabling smoother interpolation and precise control over attributes directly in the latent space. Our extensive experiments demonstrate the effectiveness of *Isometric Diffusion*, suggesting that our method helps to align latent space with perceptual semantics. This work paves the way for fine-grained data generation and manipulation.

## 1 INTRODUCTION

Generative models produce images, texts, or other types of data by learning the distribution of the observed samples in its latent space and how to map it to the actual data space. In general, we desire the latent space to reflect the human perception. That is, we wish we could find a linear subspace of the latent space that is aligned with an attribute that human perceives important to distinguish the observed samples. Equivalently, we would locate samples that look semantically similar to human nearby, and vice versa, in the latent space. Such a latent space easily disentangles the key attributes from the human's perspective, allowing us to control the generated samples as desired.

Recently, diffusion models (Sohl-Dickstein et al., 2015; Song & Ermon, 2019; Ho et al., 2020; Song et al., 2020b) have achieved unprecedented success across multiple fields, including image generation (Dhariwal & Nichol, 2021; Nichol et al., 2021; Ramesh et al., 2022; Saharia et al., 2022; Rombach et al., 2022), image editing (Kawar et al., 2023; Ruiz et al., 2023; Hertz et al., 2022), and video generation (Ho et al., 2022; Blattmann et al., 2023). However, compared to other generative models like generative adversarial network (GAN) (Goodfellow et al., 2014) or variational autoencoder (VAE) (Kingma & Welling, 2013), there are few studies exploring the latent space of diffusion models. Due to their iterative sampling process that progressively removes noise from random initial vectors, it is complicated to analyze or manipulate the latent vectors. A naive latent walking by linear interpolation between two latent vectors, for example, turns out to produce unwanted intermediate images, as illustrated in Fig. 1 (top).

A couple of recent works report important observations about the latent space $\mathcal{X}$ learned by diffusion models. First of all, Kwon et al. (2023) discovers that a diffusion model already has a semantic latent space $\mathcal{H}$ in the intermediate feature space of its score model. They suggest that $\mathcal{H}$ is semantically well-defined and locally Euclidean, and thus linear perturbations in $\mathcal{H}$ would lead to approximately linear changes in semantic attributes.

However, manipulating attributes indirectly through $\mathcal{H}$ is not fully desirable. One reason is additional computations accompanied with this indirect manipulation, as it requires two times of entire reverse diffusion process. According to Kwon et al. (2023), asymmetric reverse process is required

$x$ 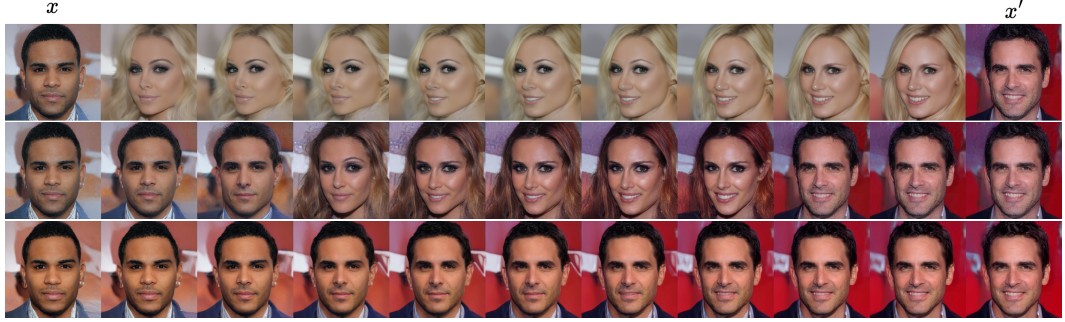 $x'$

Figure 1: **An illustration of latent traversal between two latents $x$ and $x'$** with DDPM (Ho et al., 2020), trained on $256 \times 256$ CelebA-HQ. *Top*: naive linear interpolation (Lerp) assuming Euclidean space, *Mid*: spherical interpolation (Slerp) between $x$ and $x'$ (direction $x \rightarrow x'$ is entangled with unwanted gender axis inducing abrupt changes), *Bottom*: spherical interpolation with the same latents with our Isometric Diffusion resolving unwanted entanglement.

for image change, and this requires two independent inferences of the score model with different inputs: $\epsilon_t(x_t)$ and $\tilde{\epsilon}_t(x_t) = \epsilon_t(x_t|f(x_t, t))$, where $f$ is an additional neural network to find the editing direction. Another computational cost comes from training $f$ to find local editing directions at every point of $\mathcal{H}$ for accounting every time after stepping forward in $\mathcal{H}$. With this indirect approach, a clear relationship between $\mathcal{X}$ and $\mathcal{H}$ has not been established, leaving it as an open question how to directly manipulate a particular attribute from the latent vector $x \in \mathcal{X}$ instead of $(x, h) \in \mathcal{X} \otimes \mathcal{H}$.

A subsequent work (Park et al., 2023b) suggests that a spherical linear interpolation (Slerp) in $\mathcal{X}$ is close to geodesic in $\mathcal{H}$, which implies it approximates a linear interpolation (Lerp) in $\mathcal{H}$. This discovery indicates that we may be able to manipulate semantics of a generated image directly in $\mathcal{X}$, with some care on the spherical geometry of the latent space.

To illustrate, we explore $\mathcal{X}$ by sequentially generating images on a spherically interpolated trajectory between two latent vectors, $x, x' \in \mathcal{X}$. Fig. 1 (mid) illustrates that it is not a geodesic on the data manifold; on the trajectory between two men, it unnecessarily goes through an woman. This can be interpreted that there exists some distortion in the latent space of diffusion models, implying that they fail to adequately preserve the geometric structure of the data manifold. In other words, the latent space and perceptual semantics do not align well. Such a misalignment often leads to entanglement of multiple semantic concepts, making it tricky to conduct fine-grained manipulations.

Motivated from the desire to directly align the latent space with the data manifold, we present *Isometric Diffusion*, a diffusion model equipped with isometric representation learning, where isometry is a distance preserving map between metric spaces, which also preserves geodesics. More specifically, we introduce a novel loss to encourage isometry between $\mathcal{X}$ and the data manifold. With this additional supervision, the learned $\mathcal{X}$ allows semantically disentangled geodesic traversal and smoother interpolation with less abrupt changes when navigating $\mathcal{X}$, as illustrated in Fig. 1 (bottom). We demonstrate the effectiveness of our proposed method through extensive experiments, both quantitatively and qualitatively with several widely-used metrics, on multiple datasets.

## 2 LATENT SPACE OF DIFFUSION MODELS

In this section, we briefly review the latent spaces of diffusion models and illustrate the objective to achieve a better disentangled latent space.

### 2.1 LATENT SPACE $\mathcal{X}$ OF DIFFUSION MODELS

Given an observed image space, denoted by $\mathcal{X}_0$, the forward process of diffusion models repeatedly perturbs an image $x_0 \in \mathcal{X}_0$ by $x_t = \sqrt{\bar{\alpha}_t}x_0 + \sqrt{1 - \bar{\alpha}_t}\epsilon_0$, with noise $\epsilon_0 \sim \mathcal{N}(0, I)$ for $t = 1, ..., T$ and $\bar{\alpha}_t = \prod_{i=1}^{t} \alpha_i$. These perturbed images $x_t$ construct a chain of latent spaces for $t = 1, ..., T$, and the image space at each time step $t$ is denoted by $\mathcal{X}_t$. For simplicity, we denote $\mathcal{X}_T = \mathcal{X}$. To recover the original image $x_0$ from $x_T$, diffusion models train a score model $s_\theta$ by minimizing the

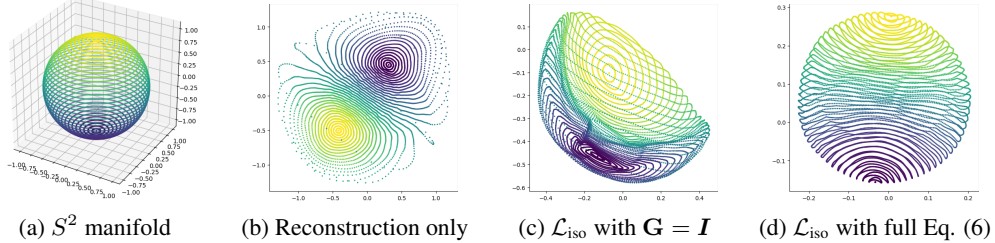

(a) $S^2$ manifold     (b) Reconstruction only     (c) $\mathcal{L}_{\text{iso}}$ with $\mathbf{G} = \boldsymbol{I}$     (d) $\mathcal{L}_{\text{iso}}$ with full Eq. (6)

Figure 2: (a) Illustration of the input $S^2$ manifold. (b–d) Mapped contours in latent coordinates learned by an autoencoder; (b) with reconstruction loss only, (c) with isometric loss assuming navie Euclidean geometry, and (d) with our isometric loss considering $S^2$ geometry.

following denoising score matching loss (Vincent, 2011; Song et al., 2020b):

$$\mathcal{L}_{\text{dsm}} = \mathbb{E}_t \left\{ \lambda(t) \mathbb{E}_{\boldsymbol{x}_0} \mathbb{E}_{\boldsymbol{x}_t | \boldsymbol{x}_0} \left[ \| \mathbf{s}_\theta(\boldsymbol{x}_t, t) - \nabla_{\boldsymbol{x}_t} \log p_t(\boldsymbol{x}_t | \boldsymbol{x}_0) \|_2^2 \right] \right\}, \tag{1}$$

where $\theta$ is a set of learnable parameters of the score model and $\lambda(t)$ is a positive weighting function.

With the trained $\mathbf{s}_\theta$, we can generate an image $\boldsymbol{x}_0$ from a sample $\boldsymbol{x}_T \sim \mathcal{N}(0, I)$ through the reverse diffusion process. Here, the distribution of the norm of completely noised images $\|\boldsymbol{x}_T\|_2$ follows a $\chi$-distribution, and they are distributed on the shell of a sphere, not uniformly within the sphere (see Sec. 3.1 for more details). For this reason, linearly interpolating two images within $\mathcal{X}$, as shown in Fig. 1 (top), results in path far from geodesic on the data manifold, while spherical linear interpolation follows a shorter path. As seen in Fig. 1 (mid), however, the spherical linear interpolation is still semantically not disentangled, indicating that $\mathcal{X}_T$ is not isometric to the data manifold.

## 2.2 Intermediate Latent Space $\mathcal{H}$ as a Semantic Space

Kwon et al. (2023) claims that the learned intermediate feature space $\mathcal{H}$ of the score model $\mathbf{s}_\theta$ sufficiently preserves the semantics of the observed images. They report that a linear scaling by $\Delta \mathbf{h}$ on $\mathcal{H}$ controls the magnitude of semantic changes, and applying the same $\Delta \mathbf{h}$ on a different sample results in a similar magnitude of effect. This implies that, by minimizing the loss in Eq. (1), $\mathcal{H}$ reasonably learns the low-dimensional data manifold with its geometry preserved and $\mathcal{H}$ is close to isometric to the data manifold. Therefore, we claim that as the mapping from $\mathcal{X}$ to $\mathcal{H}$ becomes closer to isometric, the mapping of the data manifold from $\mathcal{X}$ can also become more isometric. The advantages by achieving this objective is covered in Appendix E.

Motivated from these observations, we aim to train the encoder of the score model in a way to ensure isometry. By aligning a spherical trajectory in $\mathcal{X}$ with a geodesic in $\mathcal{H}$, our encoder paves the way for a more coherent utilization of $\mathcal{X}$ as a semantic space.

## 3 Isometric Representation Learning for Diffusion models

The goal of our work is to learn a latent space $\mathcal{X}$ which reflects semantics perceived by human. As this is not straightforward to achieve directly, we rely on a recent observation by Kwon et al. (2023) that the bottleneck layers $\mathcal{H}$ in diffusion models reasonably reflect semantics (Sec. 2.2). Thus, instead of building a semantic latent space from scratch, our approach aims to learn a geodesic-preserving mapping between $\mathcal{X}$ and $\mathcal{H}$.

For this, we claim that a scaled isometric mapping (Lee et al., 2021) guides the encoder of the diffusion model to preserve geodesics between the two spaces (Sec. 3.2), between an approximated spherical latent space $\mathcal{X}$ (Sec. 3.1) and the semantic latent space $\mathcal{H}$. Fig. 3 illustrates the overall flow of our approach. With stereographic coordinates for $\mathcal{X}$ and Cartesian coordinates for $\mathcal{H}$ as local coordinates, respectively, we equip with an appropriate Riemannian metric to the local coordinate spaces. Then, we guide the encoder of the score model to map from $\mathcal{X}$ to $\mathcal{H}$ so as to preserve geodesic between them. Lastly, we discuss computational considerations (Sec. 3.3).

**Illustration.** Before introducing our method, we first illustrate the purpose of isometric representation learning with a toy autoencoder model, learning an encoding map from $S^2$ to $\mathbb{R}^2$. The autoencoder is trained with the reconstruction loss, regularized with the isometric loss in Eq. (6).

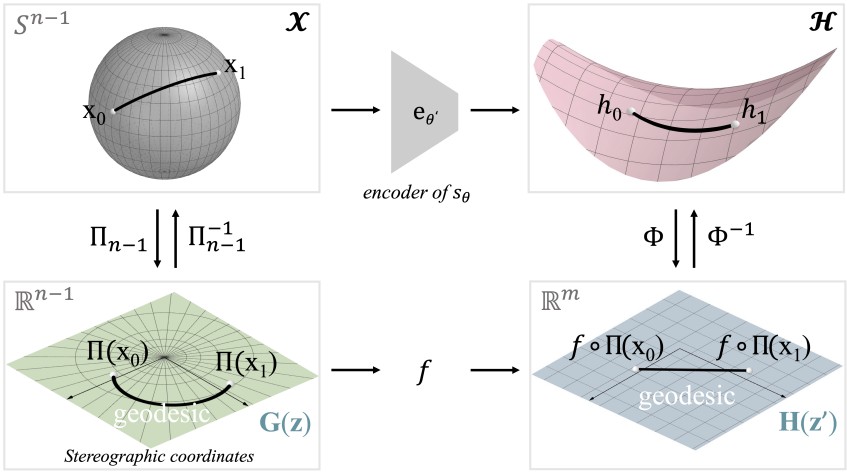

Figure 3: Illustration of $\mathcal{X}, \mathcal{H}$, and local coordinates of those two manifolds. Our isometric loss regularizes the encoder of the score model to map a spherical trajectory in $\mathcal{X}$ to a linear trajectory in $\mathcal{H}$, preserving a geodesic in $\mathcal{X}$ to a geodesic in $\mathcal{H}$. $\Pi_{n-1}, \Phi$ are charts mapping from Riemmanian manifolds to local coordinate spaces. $z, z'$ denote the local coordinates of $\mathcal{X}, \mathcal{H}$, respectively.

Fig. 2 illustrates an autoencoder flattening the given $S^2$ manifold in (a) with three different losses. Only with reconstruction loss in (b), we see that the manifold is significantly distorted, points far away in the input often are located closely. We observe less distortion with the isometric loss under the assumption of the Euclidean metric in local coordinates of $S^2$ ($\mathbf{G} = \mathbf{I}$) in (c), but it still does not preserve geodesic. With our full loss in (d), we may see that the geometry of input space is more preserved with $\mathbf{G} = \mathbf{G}_{\text{stereographic}}$ from Eq. (3). We provide more illustrations in Appendix B.

Recall that the sampling process of diffusion models starts from a Gaussian noise, $\boldsymbol{x}_T \sim \mathcal{N}(0, \boldsymbol{I}_n) \in \mathbb{R}^n$, where $T$ is the number of reverse time steps. Then, the radii of Gaussian noise vectors $\boldsymbol{x}_T$ follow $\chi$-distribution: $r = \sqrt{\sum_{i=1}^{n} \boldsymbol{x}_{T,i}^2} \sim \chi(n)$, whose mean and variance are approximately $\sqrt{n}$ and variance of 1, respectively. For a sufficiently large $n$ (e.g., $n = 3 \times 256^2$ to generate an image of size $256 \times 256$), the noise vectors reside within close proximity of a hypersphere with $r = \sqrt{n}$.

### 3.1 SPHERICAL APPROXIMATION OF THE LATENT SPACE

From this observation, we approximate the noise vectors $\boldsymbol{x} \in \mathcal{X}$ (we omit subscripts to be un-cluttered) reside on the hypersphere manifold $S^{n-1}(r) = \{\boldsymbol{x} \in \mathbb{R}^n : \|\boldsymbol{x}\| = r\}$. To define a Riemannian metric on $S^{n-1}(r)$, we need to choose charts and local coordinates to represent the Riemannian manifolds (Miranda, 1995). We choose the stereographic coordinates (Apostol, 1974) as the local coordinate to represent $\mathcal{X}$ and $\Phi = $ id following the linearity argument of $\mathcal{H}$ (Kwon et al., 2023). Stereographic projection $\Pi_{n-1} : S^{n-1}(r) \setminus \{N\} \to \mathbb{R}^{n-1}$ is a bijective transformation from every point except for the north pole $(N)$ on the hypersphere to a plane with north pole as the reference point. $\Pi_{n-1}$ and its inverse projection $\Pi_{n-1}^{-1}$ are given by

$$\Pi_{n-1}(\boldsymbol{x}) = \frac{1}{r - \boldsymbol{x}_n}(\boldsymbol{x}_1, \boldsymbol{x}_2, \cdots, \boldsymbol{x}_{n-1}), \quad \Pi_{n-1}^{-1}(\boldsymbol{z}) = \frac{r}{|\boldsymbol{z}|^2 + 1}(2\boldsymbol{z}_1, 2\boldsymbol{z}_2, \cdots, 2\boldsymbol{z}_{n-1}, |\boldsymbol{z}|^2 - 1). \quad (2)$$

In stereographic coordinates, the Riemannian metric of the $S^{n-1}(r)$ (do Carmo, 1992) is given by

$$\mathbf{G}_{\text{stereographic}}(\boldsymbol{z}) = \frac{4r^4}{(|\boldsymbol{z}|^2 + r^2)^2}\boldsymbol{I}_{n-1}, \quad \forall \boldsymbol{z} \in \mathbb{R}^{n-1}. \quad (3)$$

Recall that a diffusion model consists of a chain of latent spaces. Hence, it is needed to verify at every time step the validity of spherical approximation. From $\boldsymbol{x}_t = \sqrt{\bar{\alpha}_t}\boldsymbol{x}_0 + \sqrt{1 - \bar{\alpha}_t}\epsilon_0$, the variance of perturbation kernels is $\text{Var}[p(\boldsymbol{x}_t|\boldsymbol{x}_0)] = 1 - \bar{\alpha}_t = 1 - e^{\int -\beta(t)dt}$ (Song et al., 2020b).

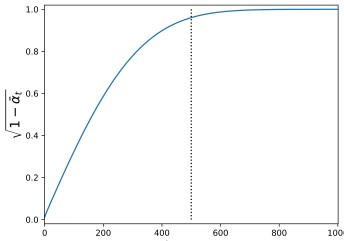

Figure 4: Scheduling of $\alpha$

We use a linear noise schedule $\beta_t = \beta_0(1 - \frac{t}{T}) + \beta_T \frac{t}{T}$ with $\beta_t = 1 - \alpha_t$, where the variance schedule is illustrated in Fig. 4. We claim that for a sufficiently large $t$, $\sqrt{1 - \bar{\alpha}_t} \approx 1$ and thus the latent space can be approximated to a sphere. That is, we approximate $\mathcal{X}_t \approx S^{n-1}(r)$ with $r = \sqrt{1 - \bar{\alpha}_t} \cdot \mathbb{E}[\chi(n)] \approx \sqrt{n(1 - \bar{\alpha}_t)}$ for $t > pT$, where we set $p \in [0, 1]$ as a hyperparamter.

## 3.2 ISOMETRIC MAPPINGS

**Definition.** An isometric mapping (or isometry) is a transformation between two metric spaces that globally preserves distances and angles. A mapping between two Riemannian manifolds $\mathbf{e}'_\theta$ : $\mathcal{M}_1 \rightarrow \mathcal{M}_2$ ($f$ in local coordinates; $f = \Phi \circ e'_\theta \circ \Pi_{n-1}^{-1}$) is a *scaled isometry* (Lee et al., 2021) if and only if

$$\mathbf{G}(\boldsymbol{z}) = c\mathbf{J}_f(\boldsymbol{z})^\top \mathbf{H}(f(\boldsymbol{z}))\mathbf{J}_f(\boldsymbol{z}), \quad \forall \boldsymbol{z} \in \mathbb{R}^{n-1}, \tag{4}$$

where $c \in \mathbb{R}$ is a constant, $\mathbf{J}_f(\boldsymbol{z}) = \frac{\partial f}{\partial \boldsymbol{z}}(\boldsymbol{z}) \in \mathbb{R}^{(n-1) \times m}$ is the Jacobian of $f$, $\mathbf{G}(\boldsymbol{z}) \in \mathbb{R}^{(n-1) \times (n-1)}$ and $\mathbf{H}(\boldsymbol{z}') \in \mathbb{R}^{m \times m}$ are the Riemannian metrics defined at the local coordinates $\boldsymbol{z}, \boldsymbol{z}'$ of $\mathcal{M}_1 = \mathbb{R}^{n-1}$ and $\mathcal{M}_2 = \mathbb{R}^m$, respectively. Equivalently, $f$ is a scaled isometry if and only if $\mathbf{J}_f^\top \mathbf{H} \mathbf{J}_f \mathbf{G}^{-1} = c\boldsymbol{I}$ where $c \in \mathbb{R}$ is a global constant. If $c = 1$ globally, $f$ is a strict isometry. Scaled isometry allows the constant $c$ to vary, preserving only the *scaled* distances and angles. This relaxation makes it easier to optimize a function to preserve geodesic with less restrictions.

In our problem formulation, $\mathcal{M}_1 = S^{n-1}$ ($\mathcal{X}$), $\mathcal{M}_2 = \mathbb{R}^m$ ($\mathcal{H}$), and $\mathbf{H}(\boldsymbol{z}') = \boldsymbol{I}_m$, as introduced in Sec. 3.1. Although evaluation of $\mathbf{J}_f^\top \mathbf{H} \mathbf{J}_f \mathbf{G}^{-1}$ is coordinate-invariant, our choice of stereographic coordinates is computationally advantageous, as its Riemannian metric in Eq. (3) is proportional to the identity matrix.

**Geodesic-preserving Property.** In order for an encoding mapping from $\mathcal{X}$ to $\mathcal{H}$ to respect the semantic structure embedded in the image space, we would like to make this mapping geodesic-preserving. We claim that the scaled isometry leads to a geodesic-preserving mapping

$$\arg\min_{\gamma(t)} \int_0^1 \sqrt{\dot{\gamma}(t)^\top \mathbf{G}(\gamma(t))\dot{\gamma}(t)}\,\mathrm{d}t = \arg\min_{\gamma(t)} \int_0^1 \sqrt{\dot{\gamma}(t)^\top \mathbf{J}(\gamma(t))^\top \mathbf{H}(f(\gamma(t)))\mathbf{J}(\gamma(t))\dot{\gamma}(t)}\,\mathrm{d}t, \tag{5}$$

for an arbitrary trajectory $\gamma : [0, 1] \rightarrow \mathbb{R}^n$ in local coordinates of $\mathcal{M}_1$ with fixed endpoints ($\gamma(0) = \boldsymbol{x}_0, \gamma(1) = \boldsymbol{x}_1$), where $\boldsymbol{x}_0, \boldsymbol{x}_1 \in \mathbb{R}^n$ are constant vectors and $\dot{\gamma}(t) = \frac{d\gamma}{dt}(t)$.

**Isometry Loss.** To sum up, we can encourage the mapping from $\mathcal{X}$ to $\mathcal{H}$ to preserve geodesics by regularizing $\mathbf{R}(\boldsymbol{z}) = \mathbf{J}_f(\boldsymbol{z})^\top \mathbf{H}(f(\boldsymbol{z}))\mathbf{J}_f(\boldsymbol{z})\mathbf{G}^{-1}(\boldsymbol{z}) = c\boldsymbol{I}$, for some $c \in \mathbb{R}$. It can be achieved by minimizing the following isometry loss:

$$\mathcal{L}_{\mathrm{iso}}(e_\theta, t) = \frac{\mathbb{E}_{\boldsymbol{x}_t \sim P(\boldsymbol{x}_t)}[\mathrm{Tr}(\mathbf{R}^2(\boldsymbol{z}_t))]}{\mathbb{E}_{\boldsymbol{x}_t \sim P(\boldsymbol{z}_t)}[\mathrm{Tr}(\mathbf{R}(\boldsymbol{z}_t))]^2} = \frac{\mathbb{E}_{\boldsymbol{x}_t \sim P(\boldsymbol{x}_t)}\mathbb{E}_{\boldsymbol{v} \sim \mathcal{N}(0,\boldsymbol{I})}[\boldsymbol{v}^\top \mathbf{R}(\boldsymbol{z}_t)^\top \mathbf{R}(\boldsymbol{z}_t)\boldsymbol{v}]}{\mathbb{E}_{\boldsymbol{x}_t \sim P(\boldsymbol{x}_t)}\mathbb{E}_{\boldsymbol{v} \sim \mathcal{N}(0,\boldsymbol{I})}[\boldsymbol{v}^\top \mathbf{R}(\boldsymbol{z}_t)\boldsymbol{v}]^2}, \tag{6}$$

where $P(\boldsymbol{x}_t)$ is the noise probability distribution at timestep $t$, and $\boldsymbol{z}_t = \Pi_{n-1}(\boldsymbol{x}_t)$. The second equality holds due to the stochastic trace estimator (Hutchinson, 1989), where $\boldsymbol{v} \in \mathbb{R}^{n-1}$ is a random vector such that $\mathbb{E}[\boldsymbol{v}\boldsymbol{v}^\top] = \boldsymbol{I}$. As a result, our final loss to train the score model is defined by

$$\mathcal{L} = \mathcal{L}_{\mathrm{dsm}} + \lambda_{\mathrm{iso}}(p, t)\mathcal{L}_{\mathrm{iso}}, \tag{7}$$

where $\lambda_{\mathrm{iso}}(p, t)$ is a non-negative weighting function to control the relative importance of isometry regularizer for each $\mathcal{X}_t$ and $p \in [0, 1]$ is the ratio of steps that we do not apply $\mathcal{L}_{\mathrm{iso}}$. We use $\lambda_{\mathrm{iso}}(p, t) = \lambda_{\mathrm{iso}}\mathbf{1}_{t'>pT}(t' = t)$ where $\mathbf{1}(\cdot)$ is the indicator function, and the denoising process starts from $t = T$.

**Applying to Diffusion Models.** The isometric loss is not directly applicable to a diffusion model, since it iteratively generates the samples. To guide a geodesic mapping between $\boldsymbol{h}_T \in \mathcal{H}$ and $\boldsymbol{x}_0$ (an actual image), we may regularize each step of the iterative sequence; that is, the encoding map between $\boldsymbol{x}_i$ and $\boldsymbol{h}_i$ for $i = 1, ..., T$.

Instead of regularizing all steps, we may selectively apply it. For time steps closer to $T$, samples are closer to a Gaussian, so our assumption may reasonably hold. For time steps closer to 0, however, samples are not sufficiently perturbed yet and thus they follow some intermediate distribution between the Gaussian and the original data distribution as described in Sec. 3.1. Hence, we may not assume these samples lie on $S^{n-1}$ manifold.

## 3.3 COMPUTATIONAL CONSIDERATIONS

To sidestep the heavy computation of full Jacobian matrices, we use stochastic trace estimator to substitute the trace of Jacobian to Jacobian-vector product (JVP). Exploiting the commutativity of the Riemmanian metric in stereographic coordinates, we utilize $\mathbb{E}_{\boldsymbol{v}\sim\mathcal{N}(0,I)}[\boldsymbol{v}^\top\mathbf{J}^\top\mathbf{J}\mathbf{G}^{-1}\boldsymbol{v}] = \mathbb{E}_{\boldsymbol{v}\sim\mathcal{N}(0,\boldsymbol{I})}[\boldsymbol{v}^\top\sqrt{\mathbf{G}^{-1}}\mathbf{J}^\top\mathbf{J}\sqrt{\mathbf{G}^{-1}}\boldsymbol{v}]$ to reduce the number of JVP evaluations. We provide more details about the computation of stochastic trace estimator in Appendix A.2.

## 4 EXPERIMENTS

We conduct extensive experiments to verify the effectiveness of our method to diffusion models and corroborate that latent space of diffusion models can be disentangled with isometric loss $\mathcal{L}_{\text{iso}}$.

## 4.1 EXPERIMENTAL SETTINGS

**Dataset.** We evaluate our approach on CIFAR-10, CelebA-HQ (Huang et al., 2018), LSUN-Church (Wang et al., 2017), and LSUN-Bedrooms (Wang et al., 2017). The training partition of each dataset consists of 50,000, 14,342, 126,227, and 3,033,042 samples, respectively. We resize each image to $256 \times 256$ except for CIFAR-10 and horizontally flip it with probability 0.5.

**Evaluation Metrics.** *Fréchet inception distance (FID)* (Heusel et al., 2017) is a widely-used metric to assess the quality of images created by a generative model by comparing the distribution of generated images with that of ground truth images. *Perceptual Path Length (PPL)* (Karras et al., 2019) evaluates how well the generator interpolates between points in the latent space, defined as PPL $= \mathbb{E}[\frac{1}{\epsilon^2}d(\boldsymbol{x}_t, \boldsymbol{x}_{t+\epsilon})]$, where $d(\cdot, \cdot)$ is a distance function. We use LPIPS (Zhang et al., 2018) distance using AlexNet (Krizhevsky et al., 2012) for $d$. A lower PPL indicates that the latent space is better disentangled, since when two or more axes are entangled and geodesic interpolation in $\mathcal{X}$ induces a sub-optimal trajectory in the semantic space, the LPIPS distance gets larger and thereby so does the PPL. For experimentation, we perform 20 and 100 steps of DDIM sampling for FID and PPL, computed with 10,000 and 50,000 images, respectively. *Linear separability (LS)* (Karras et al., 2019) measures the degree of disentanglement of a latent space, by measuring how much the latent space is separable by a hyperplane. *Mean condition number* (MCN) and *variance of Riemannian metric* (VoR) measure how much a mapping is close to a scaled-isometry, proposed by (Lee et al., 2021). We provide further details on these metrics in Appendix D.

We additionally design a new metric called *mean Relative Trajectory Length (mRTL)*, measuring the extent to which a trajectory in $\mathcal{X}$ is mapped to geodesic in $\mathcal{H}$. Specifically, mRTL is defined as the mean ratio between the L2 distance $d_2(t)$ between $\boldsymbol{h}, \boldsymbol{h}' \in \mathcal{H}$ features corresponding to two latents $\boldsymbol{x}, \boldsymbol{x}' \in \mathcal{X}$ and another distance measured on the manifold $d_{\mathcal{M}}(t)$, following along a path on $\{\mathcal{H}_t\}$. That is, RTL$(t) = \mathbb{E}_{\boldsymbol{x},\boldsymbol{x}'\in\mathcal{X}}[d_{\mathcal{M}}(t)/d_2(t)]$ and mRTL $= \mathbb{E}_t[\text{RTL}(t)]$, where $t$ denotes the timesteps of the sampling schedule. Intuitively, it represents the degree of isometry of the encoder $\mathbf{f}$.

**Implementation Details.** Our network architecture follows the backbone of DDPM (Ho et al., 2020), which uses a U-Net (Ronneberger et al., 2015) internally. We take a DDPM (Ho et al., 2020) pre-trained on CelebA (Liu et al., 2015) as a starting point, and further train it with each competing method until it achieves the lowest FID. If not specified, we train with batch size 32, learning rate $10^{-4}$, $p = 0.5$, and $\lambda_{\text{iso}} = 10^{-4}$ for 10 epochs by default. We use Adam optimizer and exponential moving average (Brown, 1956) on model parameters with a decay factor of 0.9999. We set the number of inference steps to 100. We use 4 NVIDIA A100 GPUs with 40GB memory.

## 4.2 QUANTITATIVE COMPARISON

**Overall Comparison.** In Tab. 1–2, we quantitatively compares the performance of our method and DDPM (Base) in various metrics. The results indicate that the diffusion models trained with our isometric loss regularizer exhibit substantial drop (improvement) in PPL implying smoother transitions during latent traversal. Decrease of mRTL, MCN, and VoR signified the encoder of score model became successfully closer to scaled-isometry. For CelebA-HQ, LS and LS measured by SVM with radial basis fucntion kernel significantly decreased, indicating the disentanglement of

| Dataset | FID-10k↓ | | PPL-50k↓ | | mRTL↓ | | MCN↓ | | VoR↓ | |
|---------|------|------|------|------|------|------|------|------|------|------|
| | Base | Ours | Base | Ours | Base | Ours | Base | Ours | Base | Ours |
| CIFAR-10 | **10.27** | 12.50 | 105 | **76** | 2.03 | **1.92** | 155 | **107** | **0.50** | 0.57 |
| CelebA-HQ | **15.89** | 16.18 | 648 | **570** | 2.67 | **2.50** | 497 | **180** | 1.42 | **0.85** |
| LSUN-Church | **10.56** | 13.01 | 2028 | **1587** | 3.71 | **3.21** | 375 | **217** | 1.92 | **1.37** |
| LSUN-Bedrooms | **9.49** | 11.95 | 4515 | **3809** | 3.38 | **3.21** | 320 | **186** | 1.69 | **1.12** |

Table 1: **Quantitative comparison.** Diffusion models trained with our isometric loss achieve consistent improvement over the baseline on multiple datasets, with slight sacrifice in FID scores.

| Dataset | LS ↓ | | LS (radial) ↓ | |
|---------|------|------|------|------|
| | Base | Ours | Base | Ours |
| CelebA-HQ | 4.39 | **2.65** | 12.3 | **6.8** |

Table 2: **Quantitative comparison of linear separability (LS).** LS measures the disentanglement of latent space.

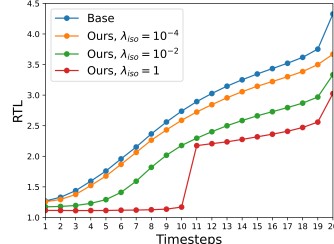

Figure 5: **RTL with various** $\lambda_{\text{iso}}$**.** A stronger regularization reduces the ratio to 1, flattening the trajectories in $\mathcal{H}$.

latent space. This further implies better alignment between the latent space and semantic space, disentangling semantic components in the latent space, as desired.

We notice a trade-off between FID and other metrics. Using our isometry loss, PPL and mRTL significantly drop, while FID sometimes marginally increases. In spite of slightly increased FID, however, the quality of the generated images is not significantly damaged, *e.g.*, as seen in examples in Fig. V. With the improved PPL and mRTL, however, latent traversal gets smoother without abrupt changes, easing controlled image manipulation (see Sec. 4.3 for more details).

**Mean Relative Trajectory Length.** Fig. 5 shows the measured Relative Trajectory Length (RTL) scores across the reverse timesteps in DDIM ($T = 20$). As the guidance of isometric loss gets larger with a larger $\lambda_{\text{iso}}$, the RTL tends to decrease, indicating the geodesic in $\mathcal{X}$ (slerp) maps to geodesic in $\{\mathcal{H}_t\}$. We notice a significant drop when $t \leq 10$ especially with a larger $\lambda_{\text{iso}}$, where the isometric loss is applied. This indeed shows the isometric loss is accurately guiding the encoder of the score model to learn an isometric representation.

## 4.3 ANALYSIS ON THE DISENTANGLEMENT OF LATENT SPACE $\mathcal{X}$

**Interpolation.** We first conduct traversals on the latent space $\mathcal{X}$ between two points $\boldsymbol{x}, \boldsymbol{x}' \in \mathcal{X}$, illustrating the generated images from interpolated points between them in Fig. 6. We observe that with our isometric loss the latent space is better disentangled, resulting in smoother transitions without abrupt changes in gender. More examples are provided in Fig. VII–VIII in Appendix H.

**Linearity.** We also claim that the latent space $\mathcal{X}$ learned with our isometric loss has a property of linearity. Specifically, we compare the generated images with ours to baseline. Both cases are naively moved along the slerp in their latent spaces. We illustrate this in Fig. 7 by demonstrating that a spherical perturbation on $\mathcal{X}$ with various intensity of $\Delta\boldsymbol{x}$ adds or removes specific attributes from the generated images accordingly. We find the editable direction by employing Local Basis (Jang et al., 2022), an unsupervised method for identifying semantic-factorizing directions in the latent space based on its local geometry, and perturb the latents through this direction both for baseline and our model. This method discovers the principal variations of the latent space in the neighborhood of the base latent code. As seen in Fig. 7, the baseline often changes multiple factors (age, gender) abruptly and inconsistently with $\gamma$ (*e.g.*, when $\gamma = -1$ on the right example, it suddenly shows a male-like output), while ours show smoother changes.

With previous diffusion-based image editing methods, one needed to take into account the geometry of $\mathcal{H}$ for every step in the editing trajectory (Park et al., 2023b). This requires computation of the Jacobian and its eigenvectors at every step forward in the trajectory via parallel transport along $\mathcal{H}$. This is usually approximated via a projection, referred as geodesic shooting. Using our isometric loss, on the other hand, the editing trajectory becomes closer to the trivial geodesic of the latent

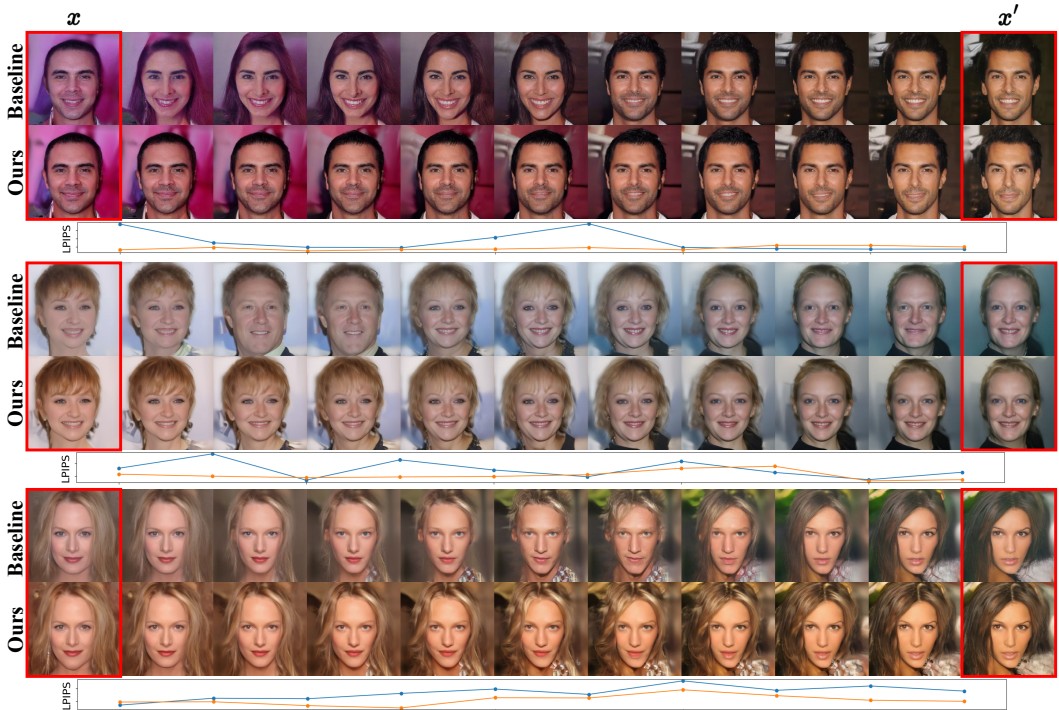

Figure 6: Examples of latent traversal between two images $x$ and $x'$ with DDPM (Ho et al., 2020), trained on $256 \times 256$ CelebA-HQ. We observe unnecessary changes of female $\rightarrow$ male in the baseline, while smoother transitions in ours. For quantitative support, we plot LPIPS distance between each adjacent frames (Blue: Baseline, Orange: Ours).

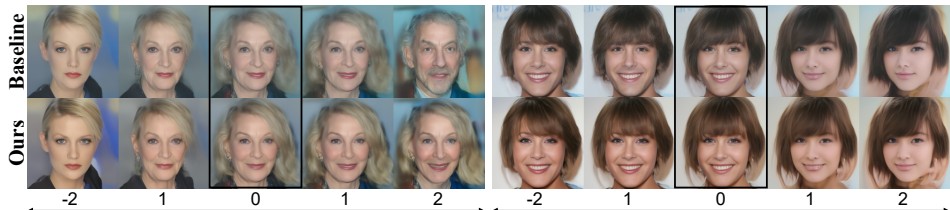

Figure 7: **Linearity.** Images generated from a latent vector $x$ (corresponds to the boxed columns) and from slightly perturbed ones, $x + \gamma \Delta x$ with $\gamma \in \{-2, -1, 0, 1, 2\}$, where $\Delta x$ corresponds to the age axis.

space; slerp in $\mathcal{X}$. Thus, we can directly move along the slerp in $\mathcal{X}$ without requiring any additional computations or approximations to find the editing direction of image.

## 4.4 ABLATION STUDY

Tab. 3 shows the ablation study on the choice of optimal $p$ and $\mathbf{G}$. With $p = 0.5$ and $\mathbf{G} = \mathbf{G}_{\text{stereographic}}$, we observe the best performance in FID and PPL. FID increases with $p < 0.5$, while PPL improvement gets marginal when $p > 0.5$. Also, when calculating the isometric loss, using an appropriate Riemannian metric $\mathbf{G}$ of the latent space turns out to be important. That is, the model with $\mathbf{G} = \mathbf{G}_{\text{stereographic}}$ achieves competitive FID and PPL scores at the same time, while either of them gets significantly worse with $\mathbf{G} = \mathbf{I}$. This result supports our spherical assumption on the latent space $\mathcal{X}$ of diffusion models and modeling it as a Riemannian manifold $S^{n-1}$ is indeed reasonable.

| $p$ | $\mathbf{G}$ | $\lambda_{\mathrm{iso}}$ | FID-10k $\downarrow$ | PPL-50k $\downarrow$ |
|---|---|---|---|---|
| 1 | - | - | 15.89 | 653 |
| 0 | $\mathbf{I}$ | $10^{-4}$ | 24.07 | 447 |
| 0.5 | $\mathbf{I}$ | $10^{-3}$ | 30.28 | 441 |
| 0.5 | $\mathbf{I}$ | $10^{-4}$ | 16.60 | 619 |
| 0.5 | $\mathbf{G}_{\mathrm{stereographic}}$ | $10^{-4}$ | 16.18 | 570 |

Table 3: **Ablation study** on $p$ (the ratio of steps to skip isometric loss) and $\mathbf{G}$ (the choice of Riemannian metric). This experiment has been conducted on CelebA-HQ $256 \times 256$.

## 5 RELATED WORKS

**Diffusion models.** Recently, diffusion models (Sohl-Dickstein et al., 2015; Song & Ermon, 2019; Song et al., 2020b) have achieved a great success in eclectic fields, containing image generation (Dhariwal & Nichol, 2021; Baranchuk et al., 2021; Choi et al., 2021b; Sehwag et al., 2022; Meng et al., 2023), image synthesis (Meng et al., 2021; Tumanyan et al., 2023; Liu et al., 2023), video generation (Ho et al., 2022; Blattmann et al., 2023) and sound generation (Yang et al., 2023). From a pure Gaussian noise, DDPM (Ho et al., 2020) samples the image by predicting the next distribution using Markov chain property. With non-Markovian process, DDIM (Song et al., 2020a) accelerates the denoising process of DDPM by skipping sampling steps.

**Latent Space of Generative Models.** On traditional Generative Adversarial Networks (GANs) (Goodfellow et al., 2014; Radford et al., 2015; Zhu et al., 2017; Choi et al., 2018; Ramesh et al., 2018; Härkönen et al., 2020; Abdal et al., 2021) models, StyleGAN (Karras et al., 2019) is a pioneering work on latent space analysis and improvement. In StyleGANv2 (Karras et al., 2020), a path length regularizer guides the generator to learn an isometric mapping from the latent space to the image space. Recently, additional studies on GANs (Shen et al., 2020a;b; Shen & Zhou, 2021) and VAEs (Hadjeres et al., 2017; Zheng & Sun, 2019; Zhou & Wei, 2020) have examined the latent spaces of generative models. Kwon et al. (2023) found that the internal feature space of U-Net in diffusion models, $\mathcal{H}$, plays the same role as a semantic latent space. Preechakul et al. (2022) discovered that using a semantic encoder enables the access to the semantic space of diffusion models. However, this method utilizes conditional diffusion model, while our work proposes a method that can directly utilize the latent space without any condition.

**Isometric Latent Space for Generative Models.** There exist some previous works on utilizing Riemannian geometry to understand the latent spaces. (Arvanitidis et al., 2021) claimed understanding Riemmanian geometry of latent space can improve analysis of representations as well as generative modeling. (Chen et al., 2020) proposed that interpreting the latent space as Riemannian manifold and regularizing the Riemannian metric to be a scaled identity help VAEs learn a good latent representation. (Lee et al., 2021) proposed an isometric regularization method for geometry-preserving latent space coordinates in scale-free and coordinate invariant form. However, due to the iterative property of diffusion models, unlike VAEs and GANs, it is demanding to apply isometric representation learning on diffusion models. Thus, to the best of our knowledge, no previous works have been done on applying an isometric mapping to the semantic space of diffusion models.

## 6 SUMMARY AND LIMITATIONS

In this paper, we have addressed a critical issue in the field of generative models, specifically unconditional diffusion models. In spite of their advances in generating photorealistic samples, they have lagged behind in terms of understanding and controlling their latent spaces.

The proposed approach, *Isometric Diffusion*, leverages isometric representation learning to bridge the gap between the latent space $\mathcal{X}$ and the data manifold. With a mapping from latent space to data manifold being close to isometry learned by our approach, we demonstrate that a more intuitive and disentangled latent space for diffusion models can be achieved both quantitatively and qualitatively.

**Limitations.** Our proposed method is applicable primarily in noise spaces close to a Gaussian distribution, limiting its applicability. Overcoming this limitation would be an interesting direction for future work.

ETHICS STATEMENT

The proposed approach in this paper aims to ease the image or video editing, selectively adjusting certain aspects of them as intended. Our work shares ethical issues of generative models that are currently known in research community; to name some, deep fake, fake news, malicious editing to manipulate evidence, and so on. We believe our work does not significantly worsen these concerns in general, but a better disentangled latent semantic space with our approach might ease these abuse cases as well. Also, other relevant ethical issues regarding potential discrimination caused by a biased dataset still remain the same with our approach, neither improving nor worsening ethical concerns in this aspect. A collective effort within the entire research community and society will be important to keep generative models beneficial.

REPRODUCIBILITY STATEMENT

We submit our code used for experiments in this paper as a supplementary material. We also plan to publicly release this upon acceptance. The readers would be able to reproduce the reported results by running this code. We also describe the detailed experimental settings including hyperparameters and hardware environments we use in Sec. 4.1 and 4.4.

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

## A  STOCHASTIC TRACE ESTIMATOR

### A.1  ESTIMATION ACCURACY

In Eq. 6 of the main text, we explained that the second quality holds because of the stochastic trace estimator (Hutchinson, 1989) which is an algorithm to obtain such an estimate from matrix-vector products:

$$\text{Tr}(A) = \mathbb{E}[\boldsymbol{v}^\top A \boldsymbol{v}] \simeq \frac{1}{N} \sum_{i=1}^{N} v_i^T A v_i, \tag{8}$$

where $A$ is any square matrix and $\boldsymbol{v}$ is random vector such that $\mathbb{E}[\boldsymbol{v}\boldsymbol{v}^\top] = \boldsymbol{I}$.

As shown in Fig. III the error of stochastic trace estimator increases as the number of sample $N$. In this experiment, $A$ follows $\mathcal{N}(0, \boldsymbol{I}) \in \mathbb{R}^{256 \times 256}$ and $\boldsymbol{v}$ follows $\mathcal{N}(0, \boldsymbol{I}) \in \mathbb{R}^{256 \times 1}$.

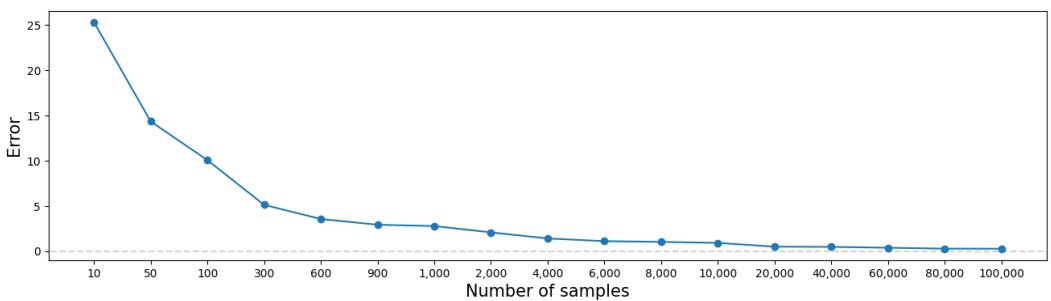

Figure I: **Approximation error of stochastic trace estimator against the number of samples.** Each point on the graph represents the error corresponding to a particular sample size.

Despite the inherent errors of estimator, we conduct a simple experiment in the setting similar to Fig. 2 to investigate whether optimizing with estimated trace converges similar to optimizing with exact trace. As shown in Fig. II, optimizing the model by approximating the trace of the matrix with the stochastic trace estimator yields similar results to those obtained by using the actual trace of the matrix. Furthermore, Fig. III demonstrates that the approximated trace exhibits a similar convergence pattern in loss over training time. These results suggest that the final convergence point is similar even when the loss function is optimized by estimating the trace of the matrix through stochastic trace estimator.

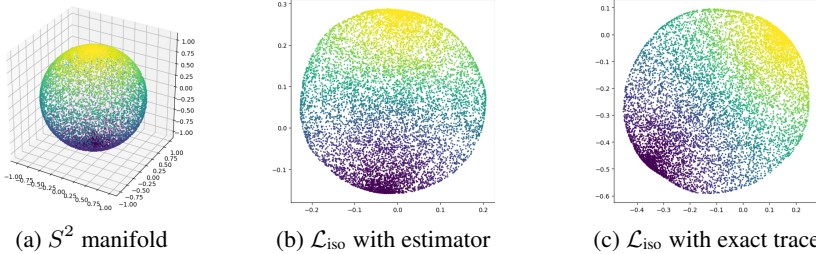

(a) $S^2$ manifold          (b) $\mathcal{L}_{\text{iso}}$ with estimator          (c) $\mathcal{L}_{\text{iso}}$ with exact trace

Figure II: (a) Illustration of the input $S^2$ manifold. (b) latent coordinates learned with isometric regularizer, estimated with the stochastic trace estimator. (c) latent coordinates learned with exact iosmetric regularizer.

### A.2  COMPUTATIONAL COMPARISON

Given that $\mathcal{X} \subset \mathbb{R}^{256 \times 256 \times 3}$ and $\mathcal{H} \subset \mathbb{R}^{8 \times 8 \times 512}$, the encoder's Jacobian, J, contains 6,442,450,944 elements. With `float32` data type, the Jacobian matrix uses approximately 24 GB of memory. The computation time for a single Jacobian takes 202.77 seconds under our environment (NVidia A100).

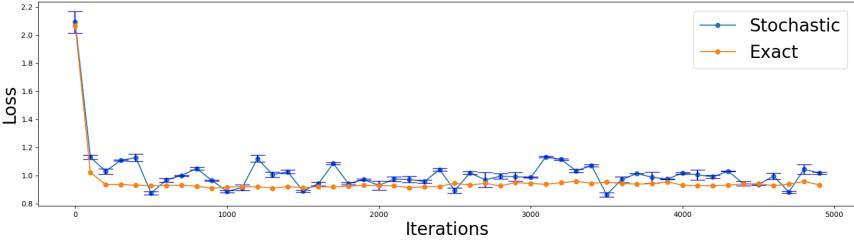

Figure III: **Loss plot during training of the toy model.** Loss calculated with trace estimator successfully converges compared to that calculated with the exact trace value. Loss calculated with trace estimator was repeated 5 times.

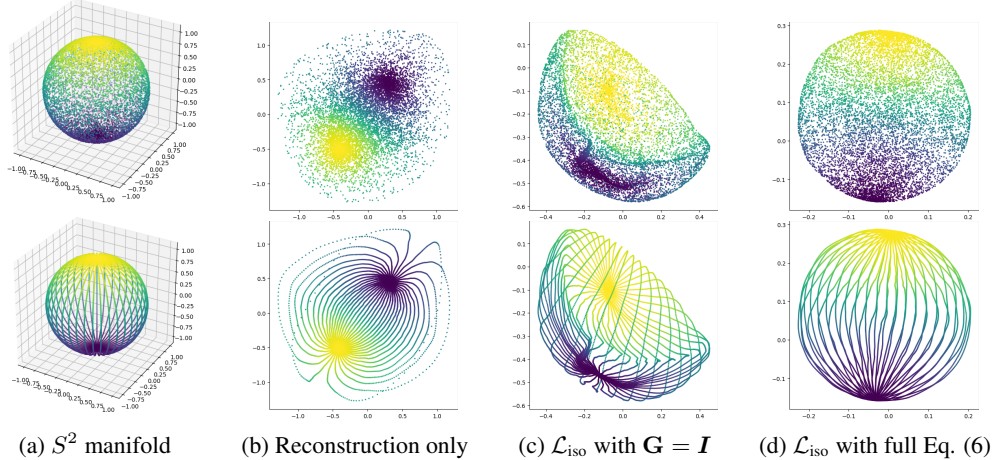

| (a) $S^2$ manifold | (b) Reconstruction only | (c) $\mathcal{L}_{\text{iso}}$ with $\mathbf{G} = \boldsymbol{I}$ | (d) $\mathcal{L}_{\text{iso}}$ with full Eq. (6) |

Figure IV: (a) Illustration of the input $S^2$ manifold. (b–d) Mapped contours in latent coordinates learned by an autoencoder; (b) with reconstruction loss only, (c) with isometric loss assuming navie Euclidean geometry, and (d) with our isometric loss considering $S^2$ geometry.

In contrast, the Jacobian Vector Product (JVP) does not explicitly calculate the entire Jacobian matrix, but it directly computes the product of the Jacobian matrix with a specific vector, requiring only $(256 \times 256 \times 3 + 8 \times 8 \times 512) \times 4 = 91{,}750$ bytes, which is approximately 0.875MB of memory. In our isometry loss, we utilize three times of JVPs for estimating a trace of J acobian. The computation time for a single JVP takes 0.6 seconds under our environment.

## B  ILLUSTRATION ON ISOMETRIC LOSS

At Fig IV, We provide more illustrations of the latent space of an autoencoder, regularized with isometric loss.

## C  LINEARITY EXAMPLES

Fig. V further illustrates the linearity of $\mathcal{X}$ with images manipulated in two directions in $\mathcal{X}$. We followed (Choi et al., 2021a) to find the editing directions. Comparing the results of baseline and ours, we observe that our method better disentangles the concept of age and gender, successfully drawing a young male and an old female (marked with red boxes), where the baseline fails to. This indicates that the latent space trained with out approach is better disentangled, and they can be easily combined back with a linear combination.

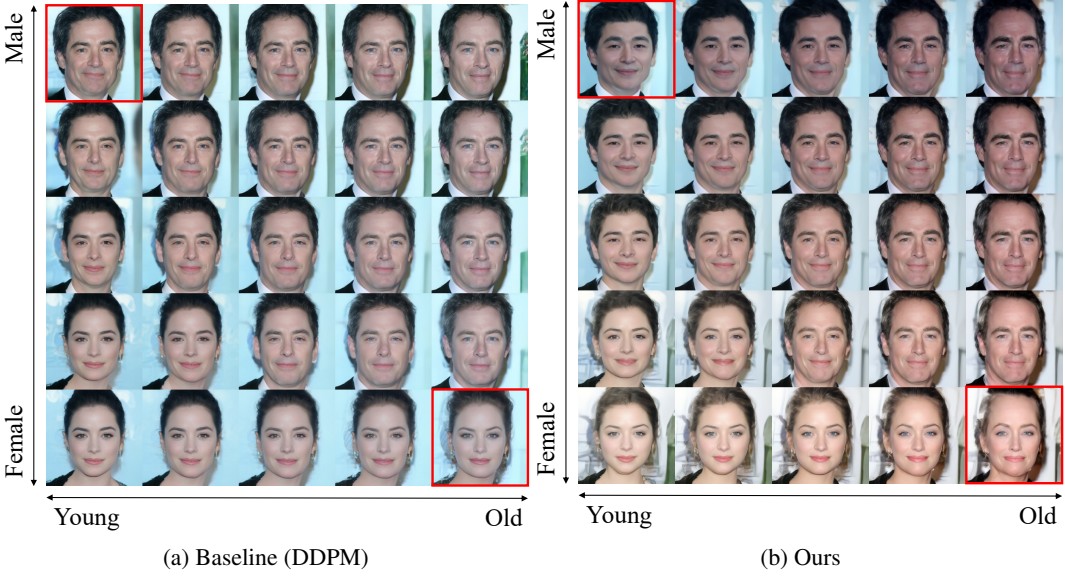

(a) Baseline (DDPM)  (b) Ours

Figure V: **An illustration of linear combination in $\mathcal{X}$.** With ours, age and gender axes (marked with red boxes) are better disentangled, and can be naturally combined back.

## D   DETAILS ON EVALUATION METRICS

We provide further details of the evaluation metrics we use throughout this paper.

*Linear separability (LS)* (Karras et al., 2019) measures the degree of disentanglement of a latent space. Karras et al. (2019) argues that if a latent space is disentangled, it should be able to find a consistent direction that changes an image attribute independently, and thus the latent space labeled according to the specific attribute should be separable by a hyperplane. The formal definition of this metric is as follows:

$$\text{LS} = e^{\sum_i H(Y_i|X_i)}, \tag{9}$$

where $i$ is the attribute index, $H(\cdot|\cdot)$ is conditional entropy, $X$ are the classes predicted by SVM, and $Y$ are the classes predicted by a pre-trained classifier. Intuitively, it measures how much additional information is needed to fully determine the label determined by the classifier, knowing the label predicted by SVM, hence indicating how much the latent space is separable by a hyperplane.

We train a classifier with ResNeXt (Xie et al., 2017) to predict the 40 attribute confidence scores with CelebA annotated for each image, and then follow the method in (Karras et al., 2019). We calculate it with SVMs using linear kernel and radial basis function kernel, regarding the spherical geometry of the latent space. We compute it with 1,000 images pruned after sorting with classifier confidence scores, from 2,000 images generated.

*Mean condition number* (MCN) and *variance of Riemannian metric* (VoR) are the metrics measuring how much a mapping is close to a scaled-isometry, proposed by (Lee et al., 2021). We measure MCN and VoR of the score models' encoders to measure how much our isometric regularizer has successfully guided the encoder to be isometric. Formally, the mean condition number (MCN) is defined as

$$\text{MCN} = \mathbb{E}_{\boldsymbol{x}_0} \mathbb{E}_{\boldsymbol{x}_t \sim p(\boldsymbol{x}_t|\boldsymbol{x}_0)} \left[ \frac{\sigma_M(\boldsymbol{J}(\boldsymbol{x}_t))}{\sigma_m(\boldsymbol{J}(\boldsymbol{x}_t))} \right], \tag{10}$$

where $\sigma_M, \sigma_m$ are the maximum and minimum singular values. MCN measures how isotropic the Riemannian metric is. Note that $\sigma_i(\boldsymbol{J}(\boldsymbol{x}_t)) = \lambda_i^2(\boldsymbol{J}^\top(\boldsymbol{x}_t)\boldsymbol{J}(\boldsymbol{x}_t))$, where $\lambda_i$ is the $i$-th eigenvalue. The variance of Riemannian metric (VoR) is defined as

$$\text{VoR} = \sum_i \text{Var}_{\boldsymbol{x}_0, \boldsymbol{x}_t \sim p(\boldsymbol{x}_t|\boldsymbol{x}_0)} \left[ \sigma_i(\boldsymbol{J}(\boldsymbol{x}_t)) \right], \tag{11}$$

where we measure how homogeneous Riemannian metric is. Note that we slightly modify its definition to bypass the exact calculation of Jacobian by exploiting SVD. Satisfying both isotropicity

| Model | FID-10k↓ | | PPL-50k↓ | | mRTL↓ | | MCN ↓ | | VoR ↓ | |
|---|---|---|---|---|---|---|---|---|---|---|
| | Base | Ours | Base | Ours | Base | Ours | Base | Ours | Base | Ours |
| DDPM | **15.89** | 16.18 | 648 | **570** | 2.67 | **2.50** | 497 | **180** | 1.42 | **0.85** |
| LDM | **10.79** | 11.46 | 439 | **397** | 2.89 | **2.73** | 322 | **198** | 1.04 | **0.54** |

Table I: **Quantitative comparison on LDM.** We trained unconditional LDM on CelebA-HQ, and compared various metrics measuring the quality of the latent space.

and homogeneity of Riemannian metric, a mapping can be determined its proximity to isometry. We measure them with 1,000 images.

## E ADVANTAGES OF SMOOTH LATENT SPACE

While there exists some topological discrepancy between Gaussian prior and the true image distribution, generative modeling have often modeled their latent spaces as Gaussian (e.g. GANs, VAEs) and there have been studies on the advantages of geometric regularizing in learning a 'better' latent space modeled as Gaussian, even though the target distribution will be quite different from it. We believe that such geodesic preserving property is motivated from various literatures in generative models.

For example, StyleGAN2 (Karras et al., 2020) uses path length regularizer to guide the generator to become closer to isometry and achieves a smoother latent space. Their work shows that the path-length-regularized StyleGAN2 improves 1) to lower PPL (a consistency and stability metric in image generation), and 2) to have invertibility from image to its latent codes. We believe the latter is potentially related to the existence of smooth inverse function of the generator, which is an important feature for image manipulation. In diffusion models, this corresponds to DDIM inversion (Dhariwal & Nichol, 2021), and we believe our method can improve the inversion quality in diffusion models and hence contribute to high quality latent manipulations, with similar effects with that of path length regularized StyleGAN2.

Additionally, FMVAE Chen et al. (2020) uses isometric regularizer to the decoder of VAE to learn a mapping from Gaussian latent space to image space close to isometry, obtaining advantages in downstream tasks using geometrically aligned latent space. As also illustrated in Karras et al. (2020); Chen et al. (2020), we admit that it somehow penalizes the FID score, possibly due to the nature of regularizer. We leave the exploration of minimizing the tradeoff as a promising future work.

Regarding the various literatures in geometric deep learning for generative modeling (Jeong et al., 2023), we would like to emphasize that our work introduces the first approach to learn a geometrically sound latent space of diffusion models which better aligns with human perceptions. Also, it advances the latent interpolation and disentanglement, which relatively has not been explored in the community.

## F REGARDING SCALABILITY OF THE METHOD

As discussed in Park et al. (2023a), the complexity of $\mathcal{H}$ increases as the complexity of the training dataset increases.

While intervention of large-scale training data, latent encoder/decoder, and text encoder in latent diffusion models (LDM)/Stable Diffusion complicates the relation between the noise space (X) and the semantic space (H), [E] demonstrates the efficacy of H space also in StableDiffusion in a text-conditioned setting, hence validating the method also in large-scale setting.

Therefore we believe the method can be scaled up, and also can incorporate text-to-image models such as StableDiffusion, which can be an interesting direction for future work. At Tab. I, we provide an additional experiment to verify our method on latent diffusion models following Dhariwal & Nichol (2021), on CelebA-HQ. As long as the H space is effective, our approach can be easily adopted to further regularize it with minimal additional cost.

# G  PRESERVATION OF $\mathcal{H}$ AFTER ISOMETRIC TRAINING

Trained with our isometric loss acting as a regularizer to the denoising score matching loss, it is not trivial if the model eventually learns the semantic space in $\mathcal{H}$. However, Kwon et al. (2023) argues that $\mathcal{H}$ exists in the bottleneck layer of the U-Net, for all pretrained diffusion models. Hence, it is reasonable to deduce that $\mathcal{H}$ space exists given that the denoising score matching (DSM) loss has converged. Therefore, it can be inferred that $\mathcal{H}$-space exists if the DSM loss converges to a similar point, even when the isometric loss is added.

We observe that the addition of the isometry loss does not significantly alter the convergence point of the diffusion loss and still shows comparable FID scores. From this, we can naturally conclude that $\mathcal{H}$-space also still exists in our model.

As empirical evidence, we provide some qualitative results of image editing with the $\mathcal{H}$ in Fig. VI. We aim to edit the image $x_0$ to the direction toward $x_0'$, manipulating only the content of the image while preserving the person's identity. Specifically, we first calculate features $\{h_t\}$ and $\{h_t'\}$ corresponding to $x_t$ and $x_t'$, respectively, where $t$ is the DDIM time steps. Then, we use $\{h_t + 1.5h_t'\}$ to inject contents during the reverse process starting from $x_T$, following Jeong et al. (2023). Note that the leftmost image for each row is $x_0$, and other images in the same row are the edited ones.

# H  LATENT TRAVERSAL EXAMPLES

We provide additional examples to compare the latent traversals with the baseline (DDPM) and with our model trained with isometric loss. Fig. VII–VIII extend Fig. 6 with more examples.

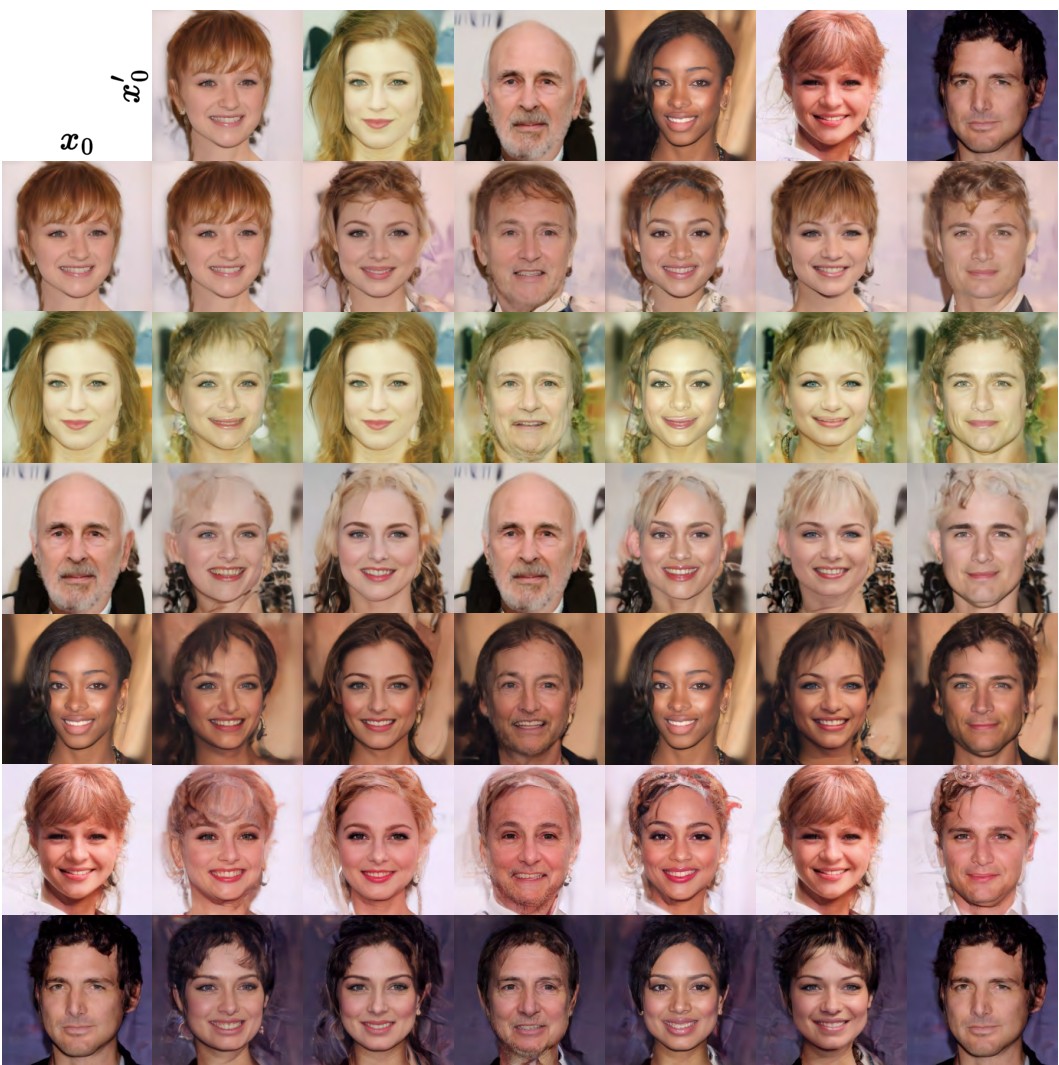

Figure VI: **Empirical observation regarding existence of $\mathcal{H}$ in our model.** Images in the same row share the original image $\boldsymbol{x}_0$, images in the same column share the source image $\boldsymbol{x}_0'$ for editing direction $\{h_t'\}$.

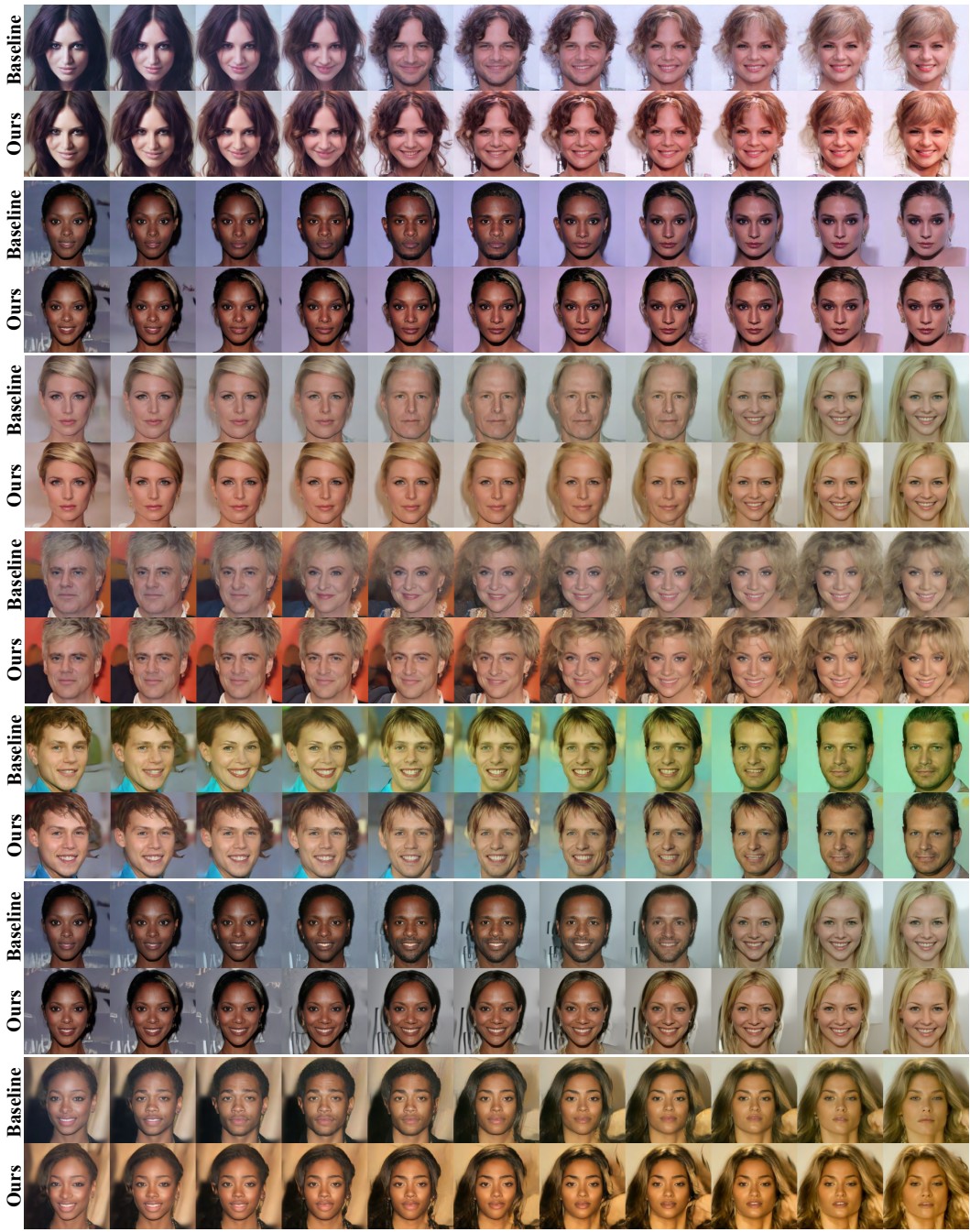

Figure VII: **Additional examples of latent traversal** between two images $x$ and $x'$ with DDPM (Ho et al., 2020) and model trained with isometric loss, trained on $256 \times 256$ CelebA-HQ.

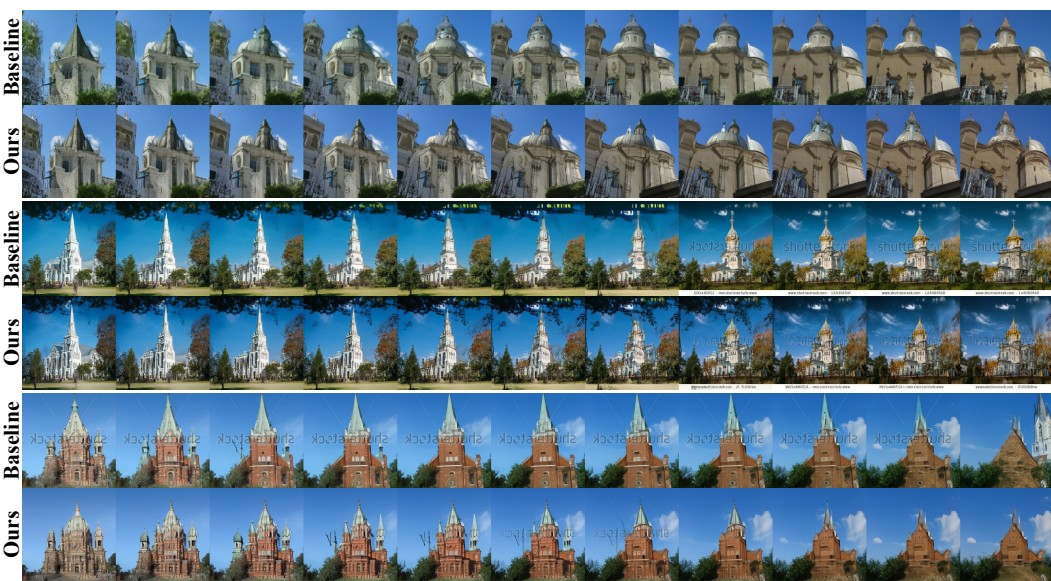

Figure VIII: Additional examples of latent traversal between two images $x$ and $x'$ with DDPM (Ho et al., 2020) and model trained with isometric loss, trained on $256 \times 256$ LSUN-Church.

