# OpenReview forum: "Isometric Representation Learning for Disentangled Latent Space of Diffusion Models"
_ICLR.cc/2024/Conference — Submitted to ICLR 2024_

### Official Review · Reviewer_foMH · 2023-10-30

**Soundness:** 4 excellent
**Presentation:** 4 excellent
**Contribution:** 3 good
**Rating:** 6
**Confidence:** 4

**Summary:**

The paper proposes to learn a representation for diffusion models, which is approximately isometric to the representation associated with the score function. The latter was found to be useful by Kwon et al (2023). The main benefit of the proposed approach is that it drastically simplifies the way in which the latent representation can be operationalised.

**Strengths:**

* The paper approaches an important problem: finding a way to leverage diffusion models for representation learning.
* The approach is quite pragmatic enabling its use in reasonably large models.
* Experiments demonstrate feasibility.
* The mathematical explanation is simple and meaningful (which is a very good thing).

**Weaknesses:**

* Several key experiments are highly subjective. For example, the opening example of Fig 1 claims that the proposed approach is better than spherical interpolation. In my subjective view, the opposite seems true. Which method actually is better is not obvious to me.
* I very much miss an early definition of 'isometry'. This is defined rather late in the paper, yet the term is used quite a bit in the first few pages to argue that the proposed approach is sensible.
* The paper several times claims that by being isometric all sorts of benefits are gained, e.g. the representation becomes disentangled and reflects semantics perceived by humans. Given that we are not talking about being isometric to a human perception space, then I struggle with the mathematical validity of such statements. I really wish the authors would tone down such claims, as the resulting paper would be much more convincing.
* A superficial reading of the paper signals that the resulting representation is an isometry. However, as far as I can tell, it is only regularized towards being an isometry. I do not object to this, but I wish the communication was more clear.
* Table 1 claims that the obtained FID scores are 'comparable' to competing methods, but, as far as I can tell, the numbers are in fact significantly worse.
* Many conclusions of the paper rely on visual inspection of interpolation plots. I honestly cannot tell which is better (e.g. I think DDPM looks better than the proposed method in Fig 8, but how am I supposed to determine which actually is better).
* There is rich literature on equipping latent representations with Riemannian geometries, see e.g. "Latent Space Oddity" by Arvanitidis et al. I miss a discussion of this in the related work as much of the tooling is the same.

All in all, while I raise many issues, they are predominantly associated with the somewhat 'overclaiming' nature of the paper writing. I do think that the proposed work has merit and is useful. Assuming the others are willing to tone down the written claims a notch, then I do support the paper.

**Questions:**

* Regarding Fig 1 it visually seems like the starting and end points are different in the three rows. Are they? If so, how do I compare?
* The 'Isometry Loss' section on page 5 proposes a specific measure. How is this different from previous isometric regularizers, e.g. the one from Lee et al.? I ask because this section reads as if it is entirely novel, whereas I suspect it might be quite similar to existing approaches for autoencoder-like models.
* The approximation in Eq 7 seems sensible. Did you test the quality of this? I would love to see some experiments showing how much is lost through this efficient approximation.
* I do not understand the illustration in Fig 4. As far as I can tell, panel d provides a highly distorted representation as the sphere is torn apart to fit the Euclidean topology (i.e. there's a massive topology mismatch between the sphere and the plane). What am I missing?
* In sec 4.4 what does "$p < 0.5$" refer to? What is $p$?
* Usually representation learning goes hand in hand with dimensionality reduction, which is not the case here. I would expect that there are fewer 'interesting' representation directions than there are image pixels (do you agree?). Does this then imply that most directions in your representation space correspond to 'weird' image changes?

---

> ### Author Response · Authors · 2023-11-20
>
> Thank you very much for your constructive and kind feedback to our work.
>
> __[W1,6. Subjective experimentation and interpretation]__
>
> We agree that evaluation of qualitative results is in nature subjective, and each person may think differently unless the quality gap is extremely obvious.
>
> To support quantitatively that our model actually generates a smoother interpolation, we added LPIPS plot between each adjacent frames at Fig. 6 (Blue: Baseline, Orange: Ours). Also, we revised Fig. 1 to present results with more pronounced differences and added Fig. 6 and VII for more examples. We hope these can provide more intuitive demonstration that smoother interpolation can be achieved for two paths sharing very similiar boundary points.
>
> Understanding the visual inspections cannot be a concrete support for our conclusion, we added several more quantitative metric comparisons supporting for how much the encoder has become close to isometry (MCN, VoR) and how much the latentspace is disentangled (LS), in Tab. 1 and 2.
>
> We hope these additional results could resolve your concerns, but we welcome further questions if you have any.
>
> __[W2. Early definition of Isometry]__
>
> Reflecting the reviewer's comment, we added an intuitive description of isometry in the introduction.
>
> __[W3. Isometry in human perception space]__
>
> We appreciate the reviewer for this constructive feedback. Our arguement that we make the mapping from human perception to latent space isometry is based on the implicit assumption that the mapping from H space to human perception is isometric, same as in [1]. However, as the reviewer pointed out, human perception space is just an abstractly defined space which can be highly subjective, and we totally agree with that the argument is not mathematically rigorous. Reflecting the reviewer's comment, we toned down our statement in the revised manuscript (marked blue). Specifically for toning down, we revised the following:
>
> - Abastrct
>   - perfectly align --> align close
>   - semantically clear and geometrically sound latent space --> semantically and geometrically better latent space
>   - marking a significant advance in aligning latent spaces with perceptual semantics. --> suggesting that our method helps to align latent space with perceptual semantics.
> - 1. Introduction
>   - each dimension in the latent space is aligned --> we could find a linear subspace of the latent space that is aligned
>   - which is known to preserve --> which is known to _better_ preserve
> - 2.2 Intermediate latent space $H$ as a semantic space
>   - Therefore, if we can make the mapping from $\mathcal{X}$ to $\mathcal{H}$ isometric, then $\mathcal{X}$ will be isometric to the data manifold $\mathcal{X}_0$ as well. --> Therefore, we claim that as the mapping from $X$ to $H$ becomes closer to isometric, the mapping of the data manifold from $X$ can also become more isometric.
>
> We have toned down the overly strong or exaggerated climas, but if you still find any parts to be excessively assertive or exaggerated, please feel free to let us know. We will gladly incorporate your feedback.
>
> [1] Kwon, Mingi, Jaeseok Jeong, and Youngjung Uh. "Diffusion models already have a semantic latent space." arXiv preprint arXiv:2210.10960 (2022).
>
> __[W4. Isometry vs. Regularizing to be isometric]__
>
> We thank the reviewer for precisely pointing this out. Yes, "regularizing towards isometry" is the most precise summary of our proposed method. Reflecting the reviewer's suggestion, we have toned down the manuscript in the revision.
>
> __[W5. Comparable FID scores]__
>
> We toned down the caption of Tab. 1, admitting that the FID slightly gets worse.
>
> __[W7. Missing literature]__
>
> We thank the reviewer for the pointer. We added "Latent Space Oddity" to the related work section.

---

> ### Author Response · Authors · 2023-11-20
>
> __[Q2. Similarity to exiting approach]__
>
> We apologize for confusion. We follow Lee at al.'s probability measure of noise space $X$. We deleted the expression in the revised manuscript.
>
> __[Q3. Loss by approximation]__
>
> Reflecting the reviewer's comment, we have added an additional experiment on the stochastic trace estimator to test its estimation quality in Appendix A. However, since we cannot get the exact Jacobian matrix because of computational issue, we conduct the experiment on a toy model. As shown in Fig. I, we firstly measure the error of the estimation against the number of samplers. Although there is a significant error initially, the error decreases as the numbers increase. It indicates that with iterative learning, our estimation approaches the true value.
>
> To demonstrate this experimentally, we compare the loss implemented with a trace estimator in a toy model to the loss implemented with the exact trace. As seen in Fig. III, we observe that the loss implemented with the trace estimator converges to a similar point as the loss implemented with the exact trace.
>
> __[Q4. Illustration in Fig. 4]__
>
> We changed Fig 4 (now it's Fig. 2 in the revision) to illustrate a more intuitive picture of mappings closer to isometry by plotting the mesh grids (geodesics) on the sphere. We provide illustration to give intuition that vanilla autoencoder induces latent coordinates that does not respect the geometry of the input space (e.g. two poles colored as yellow and purple lies closer while two neighboring green contours lie far away in panel b), indicating that the vanilla autoencoder's result is farther from exact isometry than the isometrically regularized autoencoder's.
>
> On the other hand, trained with the isometric loss, the autoencoder induces nicer latent coordinates that does not show unwanted warps (closer to isometry), despite the fact that 'exact' isometry cannot be achieved due to the mentioned natural topology mismatch. We hope that the new Fig. 2 illustrates a better intuitive explanation for isometric representations.
>
> __[Q5. What is $p$?]__
>
> $p$ denotes the ratio of steps to skip isometric loss. Specifically, we multiply $\lambda_\text{iso}(p,t) = \lambda_\text{iso}1_{t'>pT}(t'=t)$ to $\mathcal{L}_\text{iso}$ in order to regularize only for the timesteps t s.t. $500 < t \le 1000$ (p=0.5).
> We added the explanation to the timestep cutoff variable $p$ in our revised manuscript.
>
> __[Q6. Interesting directions in the dimension of image pixel space]__
>
> We agree that there exists much fewer 'interesting' representation directions than the dimension of image pixel space. After exploring the latent space by changing the latent with random directions, we were able to discover that most of the directions correspond to indescribable or indistinguishable changes. We understand that this question arises from the peculiar yet interesting feature of diffusion model; dimension of latent space (noise space) and the image space is same.
>
> If we define 'semantic space' as the space composed of axes corresponding to attributes perceivable by human (for example, the coordinates of an image is composed with strength of how much the image is close to being male, young, ...), where we assume human perception as a function from image to semantic space, the dimension of semantic space will be much smaller than the dimension of latent space (pixel space) for diffusion models. This implies that the generative process is non-injective: there exists $(v_i)_{i \in I}$ s.t. $f(v_j)=f(v_k)$ for all $j,k \in I$ where f is human perception $\circ$ generative process. As the dimensionailty is reduced significantly, $|I|$ will usually get very large. This implies most of the directions will change the 'interesting' attributes of the image, while most of the directions will correspond to change of multiple attributes, hence, the change is indistinguishable (e.g., destructive interference between multiple attributes) or indescribable (something has changed but cannot be simply described) to human perception.

---

> ### Author Response · Authors · 2023-11-20
>
> __[Q1. About starting/ending points]__
>
> The first two rows of Fig. 1 have been generated from the same baseline model sharing the boundary (starting and ending) points, while the thrid row has been generated from our model from very similiar but different boundary points. We completely agree with the reviewer's point that having identical starting and ending points is advantageous for comparison. Unfortunately, however, due to the nature of training from scratch, the weights of the two score models cannot be identical, resulting in different images at the start and end points.
>
> However, we used the exactly same initial latent (noise) for same points, and were able to obtain very similar images, maintaining high-level identities, which is presented in the figure. We argue that because the difference of score models in original diffusion model vs. in our model, that is, $s_\theta, \tilde{s}_\theta$ will be marginal (given that each desnoising score matching loss has converged), boundary points generated from exactly same latents using two score models will lie in each other's close neighborhood on the data manifold, and comparing smoothness of two paths sharing similiar boundary points is reasonable. In short, the reported images are the results of our best effort to make the comparison as fair as possible, hoping the reviewer can understand this difficulty.

---

> > ### Comment · Reviewer_foMH · 2023-11-20
> > **Thanks**
> >
> > I appreciate the changes made to the paper (thanks for highlighting them in blue for easy inspection). Indeed these fixes my main objections.

---

> > > ### Author Response · Authors · 2023-11-20
> > >
> > > Thanks for your quick response and admitting our changes.
> > > We were able to significantly improve the paper based on your feedback, and have thought about several insightful questions.
> > > Although it is a bit long, please take a look the answers and let us know if you have any further questions or additional concerns. We will be more than happy to further discuss if you have any remaining concerns.
> > > If not, we would appreciate if you can also consider raising the score as well.
> > > Thanks again for your insightful feedback and supporting our paper!

---

> > > > ### Comment · Reviewer_foMH · 2023-11-21
> > > > **Regarding the score...**
> > > >
> > > > I will revisit my score after having discussed it with fellow peer reviewers in a closed session (otherwise the scoring dynamics quickly get out of control)

---

> > > > > ### Author Response · Authors · 2023-11-21
> > > > >
> > > > > Thank you for your quick reply. Please feel free to ask us any additional question if you have.
> > > > > Thanks again for your insightful feedback and support!

---

### Official Review · Reviewer_KgHU · 2023-10-31

**Soundness:** 2 fair
**Presentation:** 2 fair
**Contribution:** 2 fair
**Rating:** 5
**Confidence:** 4

**Summary:**

Even though Diffusion Models have shown impressive performance, their latent space has remained underexplored. Thus, the authors are introducing an objective function for training Diffusion Models to make $X$ space isometric. By adding the proposed objective to the original Diffusion objective, the authors show that the $X$ space can have smoother interpolation as well as better disentanglement.

**Strengths:**

1. Interesting idea
2. An intuitive illustration of the effect of the proposed method in a simple setting is contained in the paper.

**Weaknesses:**

1. To me, it is not straightforward how “ISOMETRIC REPRESENTATION LEARNING” is related to “DISENTANGLED LATENT SPACE“.
2. It is unclear why it is needed to have $X$ space isometric while degrading the performance w.r.t. FID. It seems like the $H$ space in [1] is not degrading any performance of the original Diffusion Models since they fix the pre-trained models.
3. It seems like the $H$ space in [1] (considered a reliable semantic space by the authors) is obtained from the pre-trained Diffusion Models. If Diffusion Models are trained from scratch with an additional objective, how do the authors ensure that the $H$ space in [1] and the $H$ space in this paper have similar properties?
4. Missing baseline: since the proposed method in this paper requires training from scratch, I think [2] needs to be mentioned and compared.
5. Confusing notations; please see Questions 2,3,4, and 5.
6. The introduction is not persuasive enough about why the proposed method is needed.

[1] DIFFUSION MODELS ALREADY HAVE A SEMANTIC LATENT SPACE, Kwon et al., ICLR'23
[2] Diffusion Autoencoders: Toward a Meaningful and Decodable Representation, Preechakul et al., CVPR'22

**Questions:**

1. The reviewer understands $H$ as a pre-trained feature space of UNet (i.e., the bottleneck layer). There are several questions regarding the $H$ space that are not clear to me.
    1. The authors have mentioned in the Introduction that “However, manipulating attributes indirectly through $H$ is not fully desirable. One reason is additional computations accompanied with this indirect manipulation, as it requires two times of the score model evaluations and training an extra neural network to find local editing directions at every point of $H$.”
        1. What does it mean by “as it requires two times of the score model evaluations”? Can it be understood without knowing the paper by Kwon et al. [1]? I also read Section 2.2, but it is still not understandable. Why does the method in [1] require sampling twice? and why does not it happen in the proposed method in this paper?
        2. From this part “training an extra neural network to find local editing directions at every point of $H$”, I would assume that it means the method in [1] requires to find a semantically meaningful direction by training an additional module. Here is a question. Why does it not happen in $X$ space while it is needed in $H$ space?
2. What are the $z$ and $z’$?
3. What is the $\it{f}$ on the bottom of Fig. 2? I saw the definition “A mapping between two Riemannian manifolds”, but it is not clear how it is implemented. Also, using a different letter with the encoder of $s_\theta$ is recommended to avoid any confusion.
4. What is $\phi$? in Fig. 2?
5. What is the input of $\mathcal{L}_{iso}$?
6. I recommend moving the illustration part (Fig. 4 and its description) to the beginning of Section 3.
7. What are the interesting applications that the proposed method shows better performance than the baseline?
8. How was the baseline interpolation done? Is it based on the naive slerp? If so, from my perspective, it is mandatory to compare the results with [1] since the authors have mentioned that [1] is an important related work.
9. How is the semantic direction in Fig. 8 found? It would be much better to compare if the images on the starting/ending points are the same.
10. The authors argue their proposed methods have 1. better interpolation and 2. better disentanglement. What is the measure supporting the second argument?

---

> ### Author Response · Authors · 2023-11-19
>
> Thank you for your constructive and valuable feedbacks! We sincerely appreciated them which allowed us to revise our work.
>
> __[W1. Relation between Isometric Representation Learning and Disentangled Latet Space]__
>
> As mentioned in [B], there are various definitions for disentanglement, but a common goal in generative modeling is a latent space that consists of linear subspaces, each of which controls one factor of variation. A latent space is considered ideal if it maximally achieves this.
>
> A key assertion in [1] is that the H-space of the diffusion models are closely aligned with the semantic space. Mathematically, this can be interpreted as the mapping from the H space to the semantic space being close to isometric. Furthermore, it implies that the more isometric the encoder mapping from X-space to H-space is, the more disentangled X-space becomes. In this respect, we aim to achieve the disentanglement of the latent space by regularizing the encoder to be as isometric as possible through optimizing the isometric loss.
>
> To measure how isometric the encoder becomes, we measured mRTL in the original manuscript. Recognizing that this might not be sufficient, we additionally measure mean condition number (MCN) [A] and variance of Riemannian metric (VoR) [A] for further verification. In Tab. 1 in the revised manuscript, we see that the use of isometry loss results in lower mRTL, MCN, and VoR across all datasets, indicating that the encoder has successfully become closer to isometric.
>
> Although we have conceptually explained that achieving an isometric encoder leads to a disentangled latent space, we would like to more intuitively measure disentanglement of latent space. To this end, we adopt perceptual path length (PPL) and linear separability (LS), metrics used by StyleGAN [B] to claim the latent space is disentangled. The results in Tab. 1 and 2 show a significant reduction in these metrics, signifying that our latent space is actually disentangled.
>
>
> __[W2. Advantages of achieving isometry]__
>
> Although we observe some increase in FID, we respectfully argue that obtaining isometric mappings between latent space and data manifold (and obtaining disentangled latent space) outweighs drawbacks of some degradation in perceptual quality.
>
> First, editing in $H$ space is _computationally expensive_ regarding both training and inference. At training, to find an editing direction corresponding to an attribute like "smiling" direction, [1] needs to train an additional neural network $f(h_t, t) \in H_t$ at all timesteps $t$. On the other hand, if the geometry of $X$ and that of data manifold is mapped with minimal distortion (=close to isometry), we need to learn only a consistent vector $v \in T_{x_T}S^2$. In this way, our method may achieve significant reduction in training time. At inference, due to the nature of asymmetric reverse process, [1] needs two times of entire reverse diffusion process. As the main bottleneck of a diffusion model is this time-consuming iterative inference, this implies the entire inference cost is nearly doubled compared with ours. Also, since the mapping of geodesic of $X$ in previous diffusion models are far from geodesic in the data manifold, smooth interpolation is still not achieved with [1], while our approach achieves.
>
> Second, [1] aims to propose an image editing method, and we propose a method to learn a better (disentangled) latent space, hence the objective of each paper is different and directly comparing two methods is difficult. It is important to obtain an interpretable and disentangled latent space in generative models, essential for a proper image editing and analysis of the generative process itself [C, D]. We propose isometric representation learning to achieve disentangled latent space of diffusion models, which is advantageous in obtatining smooth interpolations and fine-grained control of the synthesis.
>
> For these reasons, we argue that guiding the diffusion model to learn a geometry-preserved latent space ($X$) is important and advantageous, and outweighs the disadvantage of minimal FID increase.
>
>
> [A] Lee et al., "Regularized autoencoders for isometric representation learning." ICLR 2021.
>
> [B] Karras, Tero, Samuli Laine, and Timo Aila. "A style-based generator architecture for generative adversarial networks." CVPR 2019.
>
> [C] Aoshima, Matsubara. "Deep Curvilinear Editing: Commutative and Nonlinear Image Manipulation for Pretrained Deep Generative Model." CVPR 2023.
>
> [D] Arvanitidis, Hansen, Hauberg. "Latent space oddity: on the curvature of deep generative models." arXiv:1710.11379 (2017).

---

> ### Author Response · Authors · 2023-11-19
>
> __[W3. Preservation of H space with our method]__
>
> [1] argues that H-space exists in the bottleneck layer of the U-Net, for all pretrained diffusion models, and it is reasonable to deduce that H space exists given that the denoising score matching (DSM) loss has converged. Therefore, it can be inferred that H-space exists if the diffusion loss converges to a similar point even when the isometric loss is added.
>
> Additionally, we can check with the FID scores to see if the diffusion loss has converged well. We observed that the addition of the isometry loss to diffusion loss does not significantly alter the convergence point of the diffusion loss and shows comparable FID scores, which leads to natural inference that H-space will also exist in our diffusion model.
>
> However, as the reviewer suggested, the introduction of an additional objective could potentially alter the property of H-space. Therefore, we have conducted an additional experiment using our model to show that our H-space has similar properties. Please see Appendix E and Fig. VI in the revision.
>
> __[W4. Missing Baseline]__
>
> We thank the reviewer for this suggestion. Yes, Diffusion Autoencoders [2] also presented a method that achieves a very smooth interpolation, sharing a similar goal with ours in some sense. The biggest difference from ours is that [2] uses a conditional diffusion model, conditioned on an additional latent variable $z_\text{sem}$, which is an output of an additional semantic encoder. Our method, on the other hand, uses an unconditional diffusion model. More specifically, Diffusion autoencoders [2] interpolates by $z(t) = (Lerp(z_{\text{sem}}^1, z_{\text{sem}}^2; t), Slerp(x_T^1, x_T^2; t))$, while we do it by a naive Slerp, $Slerp(x_T^1, x_T^2; t)$ without any additional conditions. We added this comparison in Sec. 5 in the revised manuscript.
>
> __[W5. Confusing notations]__
>
> We thank the reviewer for this constructive feedback. We changed notations and added descriptions accordingly, and answered each question below. The changes are highlighted at the paper with blue fonts.
>
>
> __[Q1-1. Sampling twice]__
>
> According to Theorem 1 in [1], asymmetric reverse process is required for image change, and this requires two independent inference of score model with different inputs: $\epsilon_t(x_t)$ and $\tilde{\epsilon_t}(x_t) = \epsilon_t(x_t|f_t)$, where $f_t$ is the additional module to find the editing direction. With our improved latent space $X$, sampling procedure is identical with that of ordinary diffusion models, e.g., DDIM reverse process, which needs just a single inference of score model. We revised our manuscript with this explanation in Introduction to be self-contained.
>
> __[Q1-2. Why we need additional NN only with H space?]__
>
> If the latent space is disentangled, we can manipulate an attribute on an image by obtaining a corresponding "consistent" direction in the latent space and moving along the direction, without any additional module inference. What we mean by consistent here is that for every latent vector $x$, we call a direction $\Delta x$ "consistent", if $x + \Delta x$ yields an expected image. However, since $X$ in the existing diffusion models is not disentangled, there exists no consistent direction, and thus the editing trajectory must change its direction every time after stepping forward at $X_t$ or $H_t$. This is the method implemented with the additional training of a neural network. In this sense, manipulating indirectly through $H$ is not fully desirable, and alternatively we aim to disentangle the $X$ space, so that we can manipulate through the consistent directions.
>
> __[Q2. $z$ and $z'$?]__
>
> $z$ and $z'$ are the local coordinates of Riemannian manifold $M_1$ and $M_2$, respectively.
> We added the description of $z$ and $z'$ in the caption of Fig. 2 (Currently Fig. 3) and in the revised main text.
>
>
> __[Q3. $f$ in Fig. 2?]__
>
> First of all, we apologize for the confusion between $f$ and $\mathbf{f}$. $f$ in Fig. 2 (Currently Fig. 3) is a map between local coordinate spaces, while $\mathbf{f}$ is a map between Riemannian manifolds, following the notation from [E].
>
> To address the confusion between notations, we changed the notation $\mathbf{f}$ to $e_\theta$ in the revision.
>
> In our notation, $f = \Phi \circ e_\theta \circ \Phi_{n-1}^{-1}$, and this is implemented by direct composition of functions. We only need $\mathbf{J}_f(z) =  \frac{\partial f}{\partial z}(z)$, so we only compute $\mathbf{J}_f(z)$, which is indirectly done utilizing jacobian vector product (JVP). Note that in our work, we use $\Phi = id.$, based on the linearity of $H$ space shown in [1].
>
> [E] Jang et al. "Geometrically regularized autoencoders for non-Euclidean data." ICLR 2022.

---

> ### Author Response · Authors · 2023-11-19
>
> __[Q4. What is $\Phi$?]__
>
> $\Phi$ in Fig. 2 (Currently Fig. 3) is a chart, mapping from Riemannian manifold to local coordinate space. [F] This could be any homeomorphism, and we choose $\Phi = \text{id}$ following the linearity argument of $\mathcal{H}$. [1]
> We added a description to the caption of Fig. 2 (Currently Fig. 3) and the main text.
>
> __[Q5. Input to $\mathcal{L}_{iso}$]__
>
> $L_{iso}$ gets the encoder of score model $e_{\theta}$, and timestep $t$. We changed the notation $L_{iso}$ to $L_{iso}(e_{\theta}, t)$ and its descriptions for clarity in Sec. 3.2.
>
> __[Q6. Moving Fig. 4]__
>
> We appreciate for this suggestion. Reflecting the reviewer's comment, we moved them to the beginning of Sec. 3 in the revised manuscript.
>
> __[Q7. Interesting applications]__
>
> First, we can perform a smooth interpolation between two images, semantically mixing them with a user-defined weight. Baseline models exhibit rough or sub-optimal interpolating results (such as appearance of woman between two men), while our work resolves this issue. In the revised manuscript, we added more interpolation results that shows our method's advantages; please see Fig. 6 and Fig. VII in the revision.
>
> Second, image editing directly in the $X$ space will be possible. As shown in Fig. 7, one may find a meaningful editing direction in the $X$ space and edit the desirable attribute by moving the $x_T$ along the direction. The $X$ space of the base diffusion models is not quite aligned with the semantic space. When $x_T$ is moved naively along a consistent direction, unexpected attributes often change together. Direct manipulation in the $X$ space is important, since it does not require access to the encapsulated pipeline (=the entire reverse process) of diffusion model.
>
> Third, Fig. V shows multiple attributes can be manipulated without dependence to each other, thanks to our disentangled latent space. This will allow easier and more precise image editing.
>
> __[Q8. Comparison with [1]?]__
>
> Yes, we use the naive slerp for both ours and the baseline interpolation.
>
> Although [1] is an important related work proposing the $H$ space as the semantic space of diffusion models, the interpolations presented in the paper is based on the _change of editing strength while $x_T$ is fixed_, whereas our interpolation is based on the _change of noise $x_T$_  itself ($x(t) = Slerp(x_T^1, x_T^2; t)$). For this reason, the comparison between ours and [1] will be not apple-to-apple.
>
> __[Q9. How we found semantic direction]__
>
> To find semantic directions, we employ Local Basis [G], an unsupervised method for identifying semantic-factorizing directions in the latent space based on its local geometry. This method discovers the principal variations of the latent space in the neighborhood of the base latent code.
>
> We completely agree with the reviewer's point that having identical starting and ending points is advantageous for comparison. Unfortunately, however, due to the nature of training from scratch, the weights of the two score models cannot be identical, resulting in different images at the start and end points. However, we used the exactly same initial latent (noise) for same points, and was able to obtain very similar images, maintaining high-level identities, which is presented at the figures.
>
> __[Q10. Metric for disentanglement]__
>
> As a response to the reviewer's question, we provide an additional metric, linear separability (LS), which has been used to measure disentanglement of the latent space by StyleGAN [B]. Please see Tab. 2 in the revised manuscript. With our method, LS significantly improves, indicating that the latent space is more disentangled.
>
> Should you have further comments or questions, feel free to share them, we would be delighted to provide answers.
>
>
>
>
> [F] Rick Miranda. Algebraic curves and Riemann surfaces, volume 5. American Mathematical Soc., 1995.
>
> [G] Choi, Jaewoong, et al. "Do not escape from the manifold: Discovering the local coordinates on the latent space of GANs." arXiv preprint arXiv:2106.06959 (2021).

---

> ### Comment · Reviewer_KgHU · 2023-11-22
>
> Thank you for the response to my review. The response clarified most of my concerns.
> I can see the efforts of the authors like reorganizing a lot of parts of the manuscript and conducting additional experiments to resolve the reviewers' concerns, which is impressive. I would say the efforts actually work, so it resolves most of my concerns.
>
> However, I still have one simple but important lingering question; "What is this for?"
>
> I saw Fig. 6 and Fig. 7 in the main paper and took a pass as well at the figures in Appendix. Even though they are interesting, I think the attribute-conditional image manipulation result (Fig. 7) is worse than Fig. 1 and Fig. 5 in [1]. I can also see that the semantic direction $\Delta x$ includes pose information which is not desirable. Second, the performance of the reference-based image translation (Fig. 6) is not very interpretable (it is not clear what attributes are transferred) and thus less practical. Maybe experiments from non-facial data (similar to [2]) can show better the usefulness of the proposed method since it is more interpretable what feature would be transferred from the reference image.
>
> Hence, I increased my initial rating to "5: marginally below the acceptance threshold", but I am still slightly leaning toward rejection because of the remaining concern about the usefulness of the proposed method in practice.
>
> Thank you again for the authors for their time and efforts for the revision.
>
>
>
> - One minor point:
>
> I guess $\theta$ for $s_\theta$ and $\theta$ for $\epsilon_\theta$ are not exactly same. If this is the case, it would be less confusing to use different notations for representing the parameters.
>
> [1] Diffusion Autoencoders: Toward a Meaningful and Decodable Representation, Preechakul et al., CVPR'22
>
> [2] DIFFUSION-BASED IMAGE TRANSLATION USING DISENTANGLED STYLE AND CONTENT REPRESENTATION, Kwon et al., ICLR'23

---

> > ### Author Response · Authors · 2023-11-23
> >
> > We thank the reviewer for the additional comments.
> >
> > __[What is this for?]__
> >
> > This work is about learning a 'better' or geometrically sound latent space of diffusion models, better aligned with human perception. In contrast to [A], which proposes an indirect manipulation in semantic space, we aim to analyze and improve the latent space by our geometric regularizer, ultimately desirable to the following aspects:
> >
> > 1) __Addressing entanglement in existing diffusion models__: In the discussion section of [C], they directly mention that "_it does not guarantee the perfect disentanglement and some directions are entangled_", and "_We have observed that some of the discovered latent vector occasionally leads to abrupt changes during the editing process_" as their limitations. Our work aims to address these issues. In fact, even a smooth latent traversal has not been possible with existing diffusion models. As seen in [E, F], latent traversal of diffusion models exhibits abrupt changes, compared to that of GANs [G]. Our method has significantly improved this, as illustrated in Fig. 1 and 6 in our revision.
> >
> > 2) __Direct image manipulation and interpolation__ without requiring access to the encapsulated diffusion pipeline. This naturally __reduces computational cost__ in image editing and interpolation to almost half, by dropping asymmetric reverse process which requires two times of entire reverse process.
> >
> > 3) Geometrically sound latent space improves image __stability and consistency__, as well as __invertibility to its latent code__. StyleGAN [D] applied the path length regularizer to achieve a similar impact on GAN, achieving a near-isometric mapping from latents to images, improving image stability and invertibility, which advances in generation quality, image manipulation. Similarly, our work regularize the mapping from X space to H space to be more isometric, and to the best of our knowledge, this is the first work demonstrating similar impacts on the diffusion model, which was previously challenging due to its iterative sampling nature.
> >
> >
> > For these reasons, we believe our work introduces a first approach to analyze and learn a 'better' latent space to the diffusion community using tools of Riemannian geometry and can also be generalized into large-scale conditional models such as StableDiffusion, supported by the demonstration given at [H].
> >
> > __[Regarding the comparison with [1]]__
> >
> > As mentioned in [1], their method is not for unconditional generation, while our method does not employ any conditional approach. For this reason, they cannot be compared apple-to-apple.
> >
> > It seems there was a misunderstanding; the term 'attribute-conditional' image manipulation by the reviewer is inappropriate for Local Basis [I] we used for Fig. 7. Local Basis is a tool to find the direction of significant variation based on a specific latent code, not to find text-conditioned semantic directions. Therefore, a fair comparison is only possible within the same model structure and using the same latent codes. We compared our method to the baseline by discovering the direction that the desired meaning is best changed among those obtained using this method.
> >
> > We believe our method can be generalized to text-conditional [2] or latent-conditional [1] methods. As [C] demonstrated the validity of H space in StableDiffusion, our method, learning a geometrically aligned X space w.r.t. the H space, is still effective. We  leave this exploration as a promising future work.
> >
> >
> > [1] Diffusion Autoencoders: Toward a Meaningful and Decodable Representation, Preechakul et al., CVPR'22
> >
> > [2] DIFFUSION-BASED IMAGE TRANSLATION USING DISENTANGLED STYLE AND CONTENT REPRESENTATION, Kwon et al., ICLR'23
> >
> > [A] Kwon, Mingi, Jaeseok Jeong, and Youngjung Uh. "Diffusion models already have a semantic latent space." arXiv preprint arXiv:2210.10960 (2022).
> >
> > [B] Song, Yang, et al. "Consistency models." (2023).
> >
> > [C] Park, Yong-Hyun, et al. "Understanding the Latent Space of Diffusion Models through the Lens of Riemannian Geometry." NeurIPS. 2023.
> >
> > [D] Karras, Tero, et al. "Analyzing and improving the image quality of stylegan." Proceedings of the IEEE/CVF conference on computer vision and pattern recognition. 2020.
> >
> > [E] https://www.wpeebles.com/DiT
> >
> > [F] https://research.nvidia.com/labs/toronto-ai/GENIE/
> >
> > [G] https://nvlabs.github.io/stylegan3/
> >
> > [H] Jeong, Jaeseok, Mingi Kwon, and Youngjung Uh. "Training-free Style Transfer Emerges from h-space in Diffusion models." arXiv preprint arXiv:2303.15403 (2023).
> >
> > [I] Choi, Jaewoong, et al. "Do not escape from the manifold: Discovering the local coordinates on the latent space of GANs." arXiv preprint arXiv:2106.06959 (2021).

---

> > ### Author Response · Authors · 2023-11-23
> >
> > __[About Fig. 6]__
> >
> > Fig. 6 is NOT an experiment to demonstrate image editing capability, but rather to show that the latent space is more disentangled compared to the baseline, by comparing two naive slerp interpolations between two latent points (x and x', we upadated Fig. 6 for clearer illustration). Hence, it is NOT a "reference-based image translation", nor a figure demonstrating an interpretable direction.
> >
> > In contrast, [2] is a well-known image translation paper solving the problem of converting a given image to a target domain. On the other hand, our work tackles a more fundamental aspect of diffusion models: learning a more disentangled latent space. This is eventually advantageous in various downstream tasks, including image editing, but we clearly mention that we are not proposing a new image editing method. For this reason, we emphasize that [2] is not a direct baseline of our work; rather, our method potentially provides a more powerful backbone diffusion model to enhance [2].
> >
> > __[Minor notation confusion]__
> >
> > We agree with the confusion. We changed $\theta \rightarrow \theta'$ for clearer notation in the revised version.
> >
> > [1] Diffusion Autoencoders: Toward a Meaningful and Decodable Representation, Preechakul et al., CVPR'22
> >
> > [2] DIFFUSION-BASED IMAGE TRANSLATION USING DISENTANGLED STYLE AND CONTENT REPRESENTATION, Kwon et al., ICLR'23

---

### Official Review · Reviewer_81Dq · 2023-10-31

**Soundness:** 4 excellent
**Presentation:** 4 excellent
**Contribution:** 4 excellent
**Rating:** 8
**Confidence:** 3

**Summary:**

This paper proposes a Isometry Loss to regularize the learning of the latent space in diffusion models such that the latent space is disentangled. The Isometry Loss is derived by enforcing the mapping between the noise space and the feature space to be an isometry using the Jacobian of the encoder and the Riemannian metrics of the two spaces. The authors did experiments on the CIFAR-10, The CelebA-HQ, LSUN-Church and the LSUN-Bedrooms datasets, and showed that the proposed method can generate realistic images and can produce smoother transitions between two images than baseline method.

**Strengths:**

1. This paper is well-motivated, and is a pioneer paper to address the disentanglement in Diffusion Models. The authors observed that the noise points in the latent space of diffusion models lie on a hypersphere, and directly performing interpolation between two noise points in the Euclidean space does not produce meaningful images. The authors proposed a solid method to address this. They introduced stereographic projection of the hypersphere and the corresponding inverse projection. They also introduced the Riemannian metrics of the stereographic projection space and the feature space. They derived the Isometry Loss by enforcing the noise space and the feature space as an isometry. The Isometry Loss is used as a regularizer for training diffusion models.

2. The proposed regularizer appear easy to use. There are only two hyperparameters: the regularization parameter and the ratio of time steps to skip the Isometric Loss. The hyperparameters are easy to tune according to ablation study.

3. The results are good. The authors first evaluate the method on an $S^2$ manifold. The authors illustrated that the method can unfold the sphere perfectly on a 2-d plane, while an autoencoder fails to. The authors evaluated the proposed method on CIFAR-10, The CelebA-HQ, LSUN-Church and the LSUN-Bedrooms datasets. The authors showed that the proposed method produced better interpolation results quantitatively and qualitatively while still maintain comparable FID scores.

**Weaknesses:**

I am not sure how heavy the computation of the Isometry Loss is. The authors used the stochastic trace estimator to substitute the trace of Jacobian to Jacobian-vector product (JVP). I am curious about how accurate the stochastic trace estimator is.

**Questions:**

I am curious about how accurate the stochastic trace estimator is.

---

> ### Author Response · Authors · 2023-11-19
>
> We sincerely thank the reviewer for valuable comments and support for our paper.
>
> __[W1. About Isometry Loss]__
>
> Given that $X \in R^{256 \times 256 \times 3}$ and $H \in R^{8 \times 8 \times 512}$, the encoder's Jacobian $J$ contains 6,442,450,944 elements. With a float32 data type, the Jacobian matrix uses approximately 24GB of memory. If we compute the Jacobian directly, it takes 202.7 seconds under our NVIDIA A100 environment.
>
> In contrast, the JVP does not explicitly calculate the entire Jacobian matrix, but it directly computes the product of the Jacobian matrix with a specific vector, requiring only $(256 \times 256 \times 3 + 8 \times 8 \times 512) \times 4 = 91,750$ bytes, which is approximately 0.875MB of memory. In our isometry loss, we compute JVP three times to estimate a trace of Jacobian. On average, computing a single JVP takes 0.6 seconds.
>
> We have added an additional experiment on the stochastic trace estimator in Appendix A based on your comments.
>
>
> Please don't hesitate to share any additional comments or questions.

---

### Official Review · Reviewer_JaPd · 2023-11-01

**Soundness:** 3 good
**Presentation:** 3 good
**Contribution:** 3 good
**Rating:** 6
**Confidence:** 3

**Summary:**

This paper proposes a method for training models based on the characteristics of the internal representations in Diffusion models. The foundation of this approach lies in the discovery that the deepest feature map of Diffusion models possesses semantically locally linear properties. The paper leverages this finding to train the model such that the Slerp (Spherical Linear Interpolation) trajectory in noise space closely approximates a geodesic in h-space, leading to a latent space that aligns more accurately with human perception. To apply the isometric loss to Diffusion, the authors successfully trained the model by utilizing assumptions applicable around T, which is close to a Gaussian distribution. In addition to the FID and PPL metrics, a new metric is introduced. Results for Unconditional Diffusion Models (DMs) across four different datasets are presented, demonstrating that this learning approach results in a latent distribution that mirrors human perceptual similarity more closely.

**Strengths:**

The approach taken in this paper is immensely intriguing. By assuming a manifold and designing a loss that aligns with this manifold, the paper provides geometric intuition through the results obtained from training the model. Complex equations are systematically explained and well-written. The paper is mathematically rigorous, and I was unable to find any significant errors.

**Weaknesses:**

I'm having trouble grasping the advantages of having a semantically linear feature in the latent space through Slerp. The benefits and points that can be gained through such a characteristic feel ambiguous and unclear.

The issue I'm noticing is that the effectiveness of this approach seems to be confined to scenarios where the noise is close to Gaussian, due to the assumptions made. Consequently, it appears that the linearity in the H-space is maintained only when the noise is Gaussian, which I find to be a kind of limitation.

Regarding the results, I'm also skeptical about how semantically linear they truly are. In Figure 1, the transition depicted is from a young man to a middle-aged man, and then to a young woman. While I can see that there is a certain degree of semantically linear progression compared to the second row, I'm not convinced that the results go beyond that. It feels like the paper could provide a more convincing demonstration of the semantically linear nature of these transformations.

**Questions:**

Could you kindly explain what is the advantage of having semantically linear feature (noise)?

**Details Of Ethics Concerns:**

Authors already provided it.

---

> ### Author Response · Authors · 2023-11-19
>
> __[W1. Advantages of semantically linear features]__
>
> We thank the reviewer for reminding this important question! The semantic linearity in our context means that as we move linearly in the latent space, the changes in the output are also linear and meaningful. For a human face, for example, we may expect moving along a straight path in the latent space would gradually change a man to a woman in the image. The key advantages here can be summarized as follows:
>
> - _Interpretability_: Semantically linear feature makes the latent space more interpretable. By observing how the output changes from perturbation in the latent space, we can understand what aspects are represented by each axis in the latent space. Without semantical linearity, perturbation in the latent space will result in abrupt or irregular changes in the output, preventing us from interpreting the latent space.
>
> - _Controllability_: Followed by interpretability, it allows direct manipulation of the images to be generated, by following the discovered direction of intended semantic changes, e.g., (anti-)aging. This interpretability and controllability is the key in the recently emerging field of diffusion-based image editing.
>
> - _Smooth Transitions_: Interpolation between two different points in the latent space becomes smoother and more continuous, preventing drastic changes in the image.
>
>
> __[W2. Non-Gaussian noise case]__
>
> The reviewer has correctly pointed out an important aspect. It would be ideal if we could ensure isometric connections from $X_t$ to $H_t$ for all timesteps $t$. However, the assumption that the latent space has a spherical geometry is feasible only when the noise is close to Gaussian, i.e., the original data distribution is perturbed with sufficent amount of noise.
>
> Hence, as demonstrated in Fig. 4, we apply a regularizer only to $t$ in [500,1000] (p = 0.5), which ensures the perturbed data distribution is sufficiently close to Gaussian, based on the $\alpha$ schedule used in the diffusion process. As shown in Tab. 3, applying the regularizer across all spaces results in a slight reduction in image quality.
>
> Advancing the regularization beyond Gaussian distribution would be an interesting direction for a future work. Precomputing the Riemannian metric of perturbed manifold corresponding to each timestep and exploiting the regularizer to all timesteps could be another direction for future work.
>
> We mention this limitation and proposed future works in the revised manuscript; please see Sec. 6.
>
>
> __[W3. Examples in Fig. 1]__
>
> We thank the reviewer for this constructive feedback. We have replaced Fig. 1 with a more intuitive example in the revised manuscript. We also provide more latent traversal examples at the revised manuscript; please refer to Fig 6 and Fig VII.
>
>
> If you have any additional comments or questions, please feel free to share them, and we will respond promptly.

---

> ### Comment · Reviewer_JaPd · 2023-11-22
>
> I would like to express my gratitude to the authors for their hard work. Also, I apologize for the late response.
>
> [Fig. 1]
> Thanks. it is now more intuitive.
>
> [Non-Gaussian noise case]
> > We mention this limitation and proposed future works in the revised manuscript; please see Sec. 6.
>
> Thank you. I think it could move into the appendix. In my opinion, this limitation is already indirectly referred to in the method/Figure.
>
>
> I have some additional questions for the authors.
>
> [Advantages of semantically linear features]
> I believe the data distribution of the real image is completely different from a Gaussian distribution. For simplicity, it can be thought of as a bimodal distribution consisting of men and women. Despite such differences in distribution, I wonder if it is a desirable property from the perspective of generative models that a certain change in the Gaussian leads to a semantic change in the image manifold.
> I guess the reason for the higher FID score is not a suitable loss for the generative model.
>
> [Scalable]
> Secondly, I am curious if authors think this training method can be scaled up to a large scale. In this paper [https://arxiv.org/abs/2307.12868], it was analyzed that as the distribution of data becomes more complex, the complexity of the H space also increases. Especially, do you think it is possible to integrate with large-scale text-to-image models like StableDiffusion?

---

> ### Author Response · Authors · 2023-11-22
>
> We sincerely thank the reviewer for prompt feedback and additional insightful questions!
>
> __[Advantages of semantically linear features]__
>
> While there exists some topological discrepancy between Gaussian prior and the true image distribution, generative modeling have often modeled their latent spaces as Gaussian (e.g. GANs, VAEs) and there have been studies on the advantages of geometric regularizing in learning a 'better' latent space modeled as Gaussian, even though the target distribution will be quite different from it. We believe that such geodesic preserving property is motivated from various literatures in generative models. Despite the fact that precomputing the topologically reasonable latent distribution may provide advantages, in some sense, the simplicity of Gaussian distribution allows easier and effective modeling and training, and this benefit may outweigh potential advantages when we try to model the distribution more precisely, even when the target distribution is not very close to it (especially since we do not explicitly know the true distribution).
>
> For example, StyleGAN2[A] uses path length regularizer to guide the generator to become closer to isometry and achieves a smoother latent space. Their work shows that the path-length-regularized StyleGAN2 improves 1) to lower PPL (a consistency and stability metric in image generation), and 2) to have invertibility from image to its latent codes. We believe the latter is potentially related to the existence of smooth inverse function of the generator, which is an important feature for image manipulation. In diffusion models, this corresponds to DDIM inversion [B], and we believe our method can improve the inversion quality in diffusion models and hence contribute to high quality latent manipulations, with similar effects with that of path length regularized StyleGAN2.
>
> Additionally, FMVAE[C] uses isometric regularizer to the decoder of VAE to learn a mapping from Gaussian latent space to image space close to isometry, obtaining advantages in downstream tasks using geometrically aligned latent space. As also illustrated in [A, C], we admit that it somehow penalizes the FID score, possibly due to the nature of regularizer. We leave the exploration of minimizing the tradeoff as a promising future work.
>
> Regarding the various literatures in geometric deep learning for generative modeling [D], we would like to emphasize that our work introduces __the first approach to learn a geometrically sound latent space of diffusion models__ which better aligns with human perceptions. Also, it advances the latent interpolation and disentanglement, which relatively has not been explored in the community.
>
>
> [A] Karras, Tero, et al. "Analyzing and improving the image quality of stylegan." Proceedings of the IEEE/CVF conference on computer vision and pattern recognition. 2020.
>
> [B] Dhariwal, Prafulla, and Alexander Nichol. "Diffusion models beat gans on image synthesis." Advances in neural information processing systems 34 (2021): 8780-8794.
>
> [C] Chen, Nutan, et al. "Learning flat latent manifolds with vaes." arXiv preprint arXiv:2002.04881 (2020).
>
> [D] Bronstein, et al. "Geometric deep learning: going beyond euclidean data.", IEEE
>
> __[Scalability]__
>
> We thank the reviewer for this great question.
>
> Yes, we believe the method can be scaled up, and also can incorporate text-to-image models such as StableDiffusion.
>
> While intervention of large-scale training data, latent encoder/decoder, and text encoder in latent diffusion models (LDM)/Stable Diffusion complicates the relation between the noise space (X) and the semantic space (H), [E] demonstrates the efficacy of H space also in StableDiffusion in a text-conditioned setting, hence validating the method also in large-scale setting. As long as the H space is effective, our approach can be easily adopted to further regularize it with minimal additional cost.
>
> We also have conducted an additional experiment to verify our method on LDMs following [F] (on CelebA-HQ; unconditioned):
>
> | Model | FID | | PPL | | mRTL | | MCN | | VoR | |
> |---|---|---|---|---|---|---|---|---|---|---|
> | | Base | Ours | Base | Ours | Base | Ours | Base | Ours | Base | Ours |
> | DDPM | __15.89__ | 16.18 | 648 | __570__ | 2.67 | __2.50__ | 497 | __180__ | 1.42 | __0.85__ |
> | LDM | __10.79__ | 11.46 | 439 | __397__ | 2.89  | __2.73__ | 322 | __198__ | 1.04 | __0.54__ |
>
> (Lower values are better for all metrics.)
>
> Although this is not text-conditioned model yet, it indicates high potential of our proposed work in the conditioned setting of LDMs [F] as well. We will update this result in the camera-ready.
>
> [E] Jeong, Jaeseok, Mingi Kwon, and Youngjung Uh. "Training-free Style Transfer Emerges from h-space in Diffusion models." arXiv preprint arXiv:2303.15403 (2023).
>
> [F] Rombach, Robin, et al. "High-resolution image synthesis with latent diffusion models." Proceedings of the IEEE/CVF conference on computer vision and pattern recognition. 2022.

---

> > ### Comment · Reviewer_JaPd · 2023-11-22
> >
> > Thanks for your quick response.
> > Most of the responses make sense to me.
> >
> > Adding this discussion will be great.
> > Especially,
> > > For example, StyleGAN2[A] uses path length regularizer to guide the generator to become closer to isometry and achieves a smoother latent space. Their work shows that the path-length-regularized StyleGAN2 improves 1) to lower PPL (a consistency and stability metric in image generation), and 2) to have invertibility from image to its latent codes. We believe the latter is potentially related to the existence of smooth inverse function of the generator, which is an important feature for image manipulation. In diffusion models, this corresponds to DDIM inversion [B], and we believe our method can improve the inversion quality in diffusion models and hence contribute to high quality latent manipulations, with similar effects with that of path length regularized StyleGAN2.
> >
> > >While intervention of large-scale training data, latent encoder/decoder, and text encoder in latent diffusion models (LDM)/Stable Diffusion complicates the relation between the noise space (X) and the semantic space (H), [E] demonstrates the efficacy of H space also in StableDiffusion in a text-conditioned setting, hence validating the method also in large-scale setting. As long as the H space is effective, our approach can be easily adopted to further regularize it with minimal additional cost.
> >
> > I like these author's responses, but of course, adding more intuition and polishing the paragraph for the paper is needed.
> >
> > Following Reviewer foMH, I'll also revisit my score after having discussed it with fellow peer reviewers in a closed session.

---

> > > ### Author Response · Authors · 2023-11-23
> > >
> > > We thank the reviewer for prompt response. Reflecting the reviewer's comment, we added the discussion about advantages of latent space equipped with linear semantic features, and the scalability of our method in Sec 2.2 and Appendix E, F. We will further revise it in the final version.

---

### Author Response · Authors · 2023-11-23
**Final remarks by authors**

We sincerely appreciate all reviewers for their insightful feedback and help to improve our paper.

__Summary of author-reviewer discussions__

We summarize the main concerns raised in the initiral review and how we addressed:

1. advantages and demonstration of semantically linear features & scalability (JaPd,KgHU): We provided additional arguments in the rebuttal and JaPd liked the answer.
2. overclaiming expressions and confusing notations (foMH): We toned down the manuscript and changed confusing notations, which fixed their main objections.
3. subjectivity in experimental results (foMH): We provided additional quantitative metrics.
4. preservation of H space with our method (KgHU): We provided additional argument and experiments, and KgHU mentioned most of the concerns were resolved.
5. purpose of acheiving disentangled latent space (KgHU): This is the only remaining issue with KgHU according to their last comment. We provided additional answers with clarification on our main purpose, contributions, and limitations.

__Summary of our contributions__

Here, we sum up the purpose of our work and contributions. Overall, we propose for the first time to learn a geometrically aligned latent space of diffusion models to human perception, by employing a geometric regularizer to guide the mapping from X space to H space closer to isometry.

1) Our work addressed entangled latent space in diffusion models, reducing abrupt changes on latent space traversal.

2) Our work allows direct image manipulation and interpolation, reducing computational cost in image editing and interpolation.

3) Geometrically sound latent space improves image stability and consistency, as well as invertibility to its latent code.

---

### Meta-Review · Area_Chair_nT4b · 2023-12-06

**Metareview:**

The paper proposed a method for learning disentangled latent space in diffusion models. In particular, it introduces an additional loss to promote isometry between the latent space and data manifold during training. Empirical results suggest that the proposed method can learn a latent space with clearer semantics. Unconditionally, it can transfer smoothly between samples.

This paper has divergent scores. As pointed out by Reviewer KgHU and agreed by the Reviewer JaPd and foMH during the reviewer discussion phase, the paper lacks a clear purpose and empirical usage. Given the rapid development of diffusion models, in particular extensive controllable methods for large-scale diffusion models, the applications (including the potential applications) considered in this paper are quite limited. In addition, it requires additional training and may slightly hurt sample quality.

I tend to reject the paper, but I will not be upset if it gets accepted.

**Justification For Why Not Higher Score:**

This paper has divergent scores. As pointed out by Reviewer KgHU and agreed by the Reviewer JaPd and foMH during the reviewer discussion phase, the paper lacks a clear purpose and empirical usage.

**Justification For Why Not Lower Score:**

N/A

---

### Decision · Program_Chairs · 2024-01-16

Reject